# Mitochondrial fatty acid synthesis coordinates oxidative metabolism in mammalian mitochondria

Sara M Nowinski[1], Ashley Solmonson[2], Scott F Rusin[3], J Alan Maschek[4,5,6], Claire L Bensard[1], Sarah Fogarty[1,7], Mi-Young Jeong[1], Sandra Lettlova[1], Jordan A Berg[1], Jeffrey T Morgan[1,7], Yeyun Ouyang[1], Bradley C Naylor[6], Joao A Paulo[3], Katsuhiko Funai[4], James E Cox[1,4,6], Steven P Gygi[3], Dennis R Winge[1,4,8], Ralph J DeBerardinis[2,7], Jared Rutter[1,4,7]*

[1]Department of Biochemistry, Salt Lake City, United States; [2]Children's Medical Center Research Institute, University of Texas Southwestern Medical Center, Dallas, United States; [3]Department of Cell Biology, Harvard University School of Medicine, Boston, United States; [4]Diabetes & Metabolism Research Center, Salt Lake City, United States; [5]Department of Nutrition and Integrative Physiology, Salt Lake City, United States; [6]Metabolomics, Proteomics and Mass Spectrometry Core Research Facilities University of Utah, Salt Lake City, United States; [7]Howard Hughes Medical Institute, Salt Lake City, United States; [8]Department of Internal Medicine, Salt Lake City, United States

*For correspondence: rutter@biochem.utah.edu

**Abstract** Cells harbor two systems for fatty acid synthesis, one in the cytoplasm (catalyzed by fatty acid synthase, FASN) and one in the mitochondria (mtFAS). In contrast to FASN, mtFAS is poorly characterized, especially in higher eukaryotes, with the major product(s), metabolic roles, and cellular function(s) being essentially unknown. Here we show that hypomorphic mtFAS mutant mouse skeletal myoblast cell lines display a severe loss of electron transport chain (ETC) complexes and exhibit compensatory metabolic activities including reductive carboxylation. This effect on ETC complexes appears to be independent of protein lipoylation, the best characterized function of mtFAS, as mutants lacking lipoylation have an intact ETC. Finally, mtFAS impairment blocks the differentiation of skeletal myoblasts in vitro. Together, these data suggest that ETC activity in mammals is profoundly controlled by mtFAS function, thereby connecting anabolic fatty acid synthesis with the oxidation of carbon fuels.

## Introduction

Fatty acids play diverse cellular roles, including providing the hydrophobic tails of membrane phospholipids, energy storage in the form of triglycerides, and as cell signaling molecules. Aside from the import of exogenous fatty acids, mammalian cells use the well-known and well-studied cytoplasmic enzyme fatty acid synthase (FASN) to make palmitate, which is further modified to form the diverse array of cellular fatty acids (*Smith, 1994*). However, it is much less appreciated that mitochondria also harbor a spatially and genetically distinct fatty acid synthesis pathway (mtFAS) (reviewed in *Nowinski et al., 2018*). In contrast to FASN, which is a very large protein that contains several domains and encompasses all of the enzymatic activities necessary for FAS condensed in a single polypeptide chain, the mtFAS pathway is comprised of at least six enzymes all encoded by separate genes. These enzymes catalyze sequential steps to achieve one cycle of two-carbon addition to a growing acyl chain (*Figure 1A*). The nascent fatty acids are covalently attached to the

**eLife digest** In human, plant and other eukaryotic cells, fats are an important source of energy and also play many other roles including waterproofing, thermal insulation and energy storage. Eukaryotic cells have two systems that make the building blocks of fats (known as fatty acids) and one of these systems, called the mtFAS pathway, operates in small compartments known as mitochondria. This pathway only has one known product, a small fat molecule called lipoic acid, which mitochondria attach to several enzymes to allow them to work properly.

The main role of mitochondria is to break down fats and other molecules to release chemical energy that powers many processes in cells. They achieve this using large groups of proteins known as ETC complexes. To build these complexes, families of proteins known as ETC assembly factors carefully coordinate the assembly of many proteins and small molecules into specific structures. However, it remains unclear precisely how this process works.

Here, Nowinski et al. used a gene editing technique to mutate the genes encoding three enzymes in the mtFAS pathway in mammalian cells. The experiments found that the mutant cells had fewer ETC complexes and seemed to be less able to break down fats and other molecules than 'normal' cells. Furthermore, a family of ETC assembly factors were less stable in the mutant cells. These findings suggest that the mtFAS pathway controls how mitochondria assemble ETC complexes. Further experiments indicated that lipoic acid is not involved in the assembly of ETC complexes and that the mtFAS pathway produces another, as yet unidentified, product that regulates this process, instead.

MEPAN syndrome is a rare neurological disorder that leads to progressive loss of control of movement, slurred speech and impaired vision in children. Patients with this syndrome have genetic mutations affecting components of the mtFAS pathway, therefore, a better understanding of how the pathway works may help researchers develop new treatments in the future. More broadly, these findings will have important ramifications for many other situations in which the activity of ETC complexes in mitochondria is modified.

mitochondrial acyl carrier protein (ACP), which acts as a soluble scaffold upon which the acyl chains are built. In each cycle of the pathway, malonyl-CoA is converted to malonyl-ACP, which then undergoes a condensation reaction with the growing fatty acyl chain on ACP, extending the chain by two carbons and releasing a $CO_2$ molecule. Thereafter, the remaining enzymes in the pathway must carry out a series of reduction and dehydration reactions to fully reduce the acyl chain to a saturated fatty acid, which is the substrate for further cycles of two-carbon addition. Among the several differences between mtFAS and cytoplasmic FAS, FASN exclusively produces palmitate, whereas mtFAS appears to have at least two major products.

Although the human genes responsible for each step in the mtFAS pathway have been identified, and their ability to complement the orthologous mutants in yeast has been demonstrated (*Autio et al., 2008*; *Chen et al., 2009*; *Joshi et al., 2003*; *Miinalainen et al., 2003*; *Zhang et al., 2005*; *Zhang et al., 2003*), few loss-of-function studies have examined the consequences of mtFAS deficiency in mammalian systems. Inducible knockout of the mitochondrial malonyl CoA-acyl carrier protein transacylase (*Mcat*) using a Cre driver that expresses in most tissues in mice results in a severe phenotype characterized by weight loss, reduced muscle strength, and shortened lifespan despite the persistence of residual MCAT protein (*Smith et al., 2012*). Knockout of the mitochondrial 2-enoyl thioester reductase, *Mecr*, is lethal in mice due to a placental defect (*Nair et al., 2017*), while the inducible knockout of *Mecr* specifically in Purkinje cells leads to loss of this cell population and recapitulates many phenotypes of MePaN syndrome, the human disease caused by *Mecr* mutation (*Gorukmez et al., 2019*; *Heimer et al., 2016*; *Nair et al., 2018*). Similarly, *Mecr* knockdown has been shown to inhibit the growth of hepatocellular carcinoma cells (*Cai et al., 2019*). While these detrimental phenotypes are clear, the molecular consequences of mtFAS loss remain poorly understood.

MtFAS currently has one known product: an eight-carbon saturated fatty acid, octanoate, that is subsequently converted to lipoic acid. This important cofactor is required for the catalytic activity of a number of mitochondrial enzymes, most notably pyruvate dehydrogenase and α-ketoglutarate

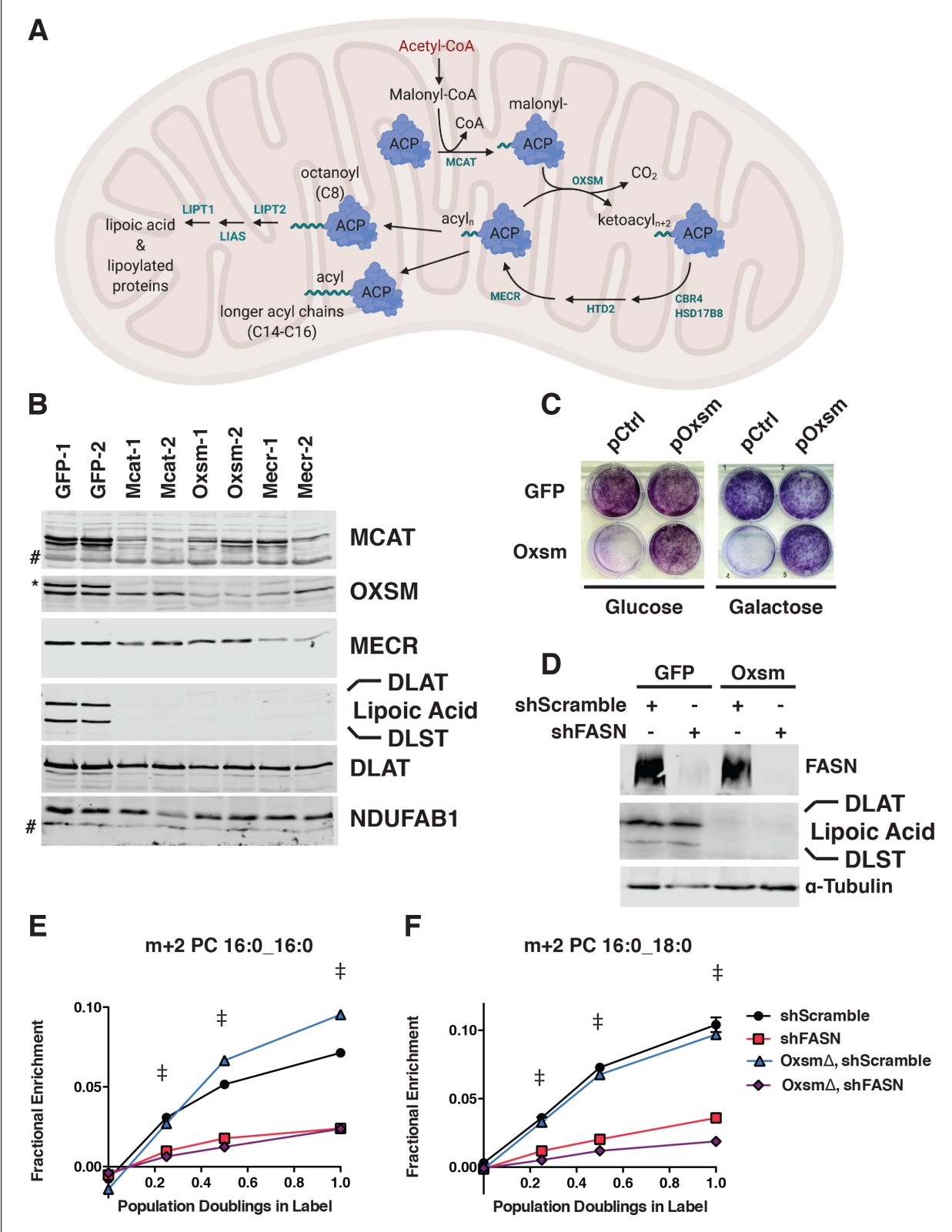

**Figure 1.** MtFAS is an essential pathway in mammalian skeletal myoblasts but does not contribute to synthesis of major cellular lipids. (**A**) Schematic of the mitochondrial fatty acid synthesis pathway and downstream lipoic acid synthesis. (**B**) Crude isolated mitochondrial fractions from duplicate single cell clones of *Mcat*, *Oxsm*, and *Mecr* mutants, compared with GFP control clonal cell lines, were separated via SDS-PAGE and immunoblotted for the indicated targets. *=Lipoic acid band (reprobe of earlier blot). #=non specific bands **C**. GFP control and *Oxsm* mutant cells were infected with retroviral
*Figure 1 continued on next page*

*Figure 1 continued*

control plasmid (pCtrl) or a plasmid expressing Oxsm off the CMV promoter (pOxsm), plated at equal densities in normal growth medium with either 4.5 g/L glucose or 10 mM galactose and grown for 3 or 4 days, respectively, then stained with crystal violet. (D) Whole cell lysates from stable cell lines generated by infecting *Oxsm* mutant cells (OxsmΔ) or GFP controls with shRNA constructs targeting *FASN* (shFASN) or scramble control (shScramble) were separated by SDS-PAGE and immunoblotted for FASN, lipoylated proteins, or tubulin. (E-F) Stable cell lines created in (D) were incubated with U$^{13}$C-glucose for the indicated number of doublings, harvested, lipids extracted, and analyzed via LC-MS. Shown are quantitation of m+2 isotopologues for two representative phospholipid species, PC 16:0_16:0 and PC 16:0_18:0. †=p < 0.001, ‡=p < 0.0001, error bars are SEM. The online version of this article includes the following figure supplement(s) for figure 1:

**Figure supplement 1.** mtFAS is required for growth of cultured skeletal myoblasts but does not significantly contribute to cellular fatty acids.

dehydrogenase (*Brody et al., 1997*; *Wada et al., 1997*), but also branched chain amino acid dehydrogenase, the H protein of the glycine cleavage system, and 2-oxoadipate dehydrogenase (reviewed in *Solmonson and DeBerardinis, 2018*). This observation, made more than 20 years ago, has guided mtFAS-focused research for the past two decades. Studies have shown that loss of lipoic acid synthesis and/or failure to efficiently transfer lipoic acid to its target proteins is lethal in mice (*Ni et al., 2019*; *Yi and Maeda, 2005*), and attributed other mitochondrial changes as downstream of lipoic acid synthesis (*Smith et al., 2012*). However, though the mouse studies described above have demonstrated striking phenotypes that result from loss of mtFAS, they fail to distinguish between direct effects of loss of protein lipoylation versus loss of other mtFAS function(s). Although lipoic acid unmistakably has important central functions in mitochondrial metabolism, there is reason to be skeptical that the sole function of mtFAS is the production of lipoic acid.

In addition to the eight-carbon precursor for lipoic acid, it is clear that mtFAS also produces longer acyl chains of at least 14 carbons, yet the identity of these lipids and their cellular functions are uncertain (*Angerer et al., 2017*; *Witkowski et al., 2007*). Perhaps more surprisingly, these longer acyl chains appear to partially be maintained in attachment to ACP (*Angerer et al., 2017*), further complicating the question of their cellular functions. We set out to systematically examine the cellular functions of the mtFAS pathway by mutating genes that encode three distinct steps in the pathway, *Mcat*, *Oxsm*, and *Mecr*. For comparison, we also engineered cells with loss of *Lipt1*, the terminal enzyme in the production of lipoylated enzymes downstream of mtFAS. We found that several phenotypes resulting from loss of the mtFAS pathway are not related to lipoic acid, but must instead be due to loss of other products or functions of mtFAS. In particular, we find that loss of mtFAS, but not lipoic acid synthesis, leads to a profound impairment in the assembly of the mitochondrial electron transport chain machinery, with the attendant consequences on metabolism and cell behaviors. Our data therefore suggest that the mtFAS pathway acts as a key regulator of mitochondrial respiratory metabolism.

## Results

### Mitochondrial fatty acid synthesis is an essential pathway that does not contribute to synthesis of cellular lipids

The enzymes of mtFAS are ubiquitously expressed in mammalian tissues, with the highest expression in skeletal muscle and heart (*Triepels et al., 1999*). Therefore, we chose cultured skeletal myoblasts as a model system in which to examine the role of this pathway. We employed a CRISPR/Cas9-based strategy to mutate three genes encoding enzymes in the mtFAS pathway, using two different guide RNA sequences per gene. We targeted *Mcat*, encoding the malonyl-CoA ACP transacylase, *Oxsm*, encoding the beta-ketoacyl synthase that condenses malonyl-ACP with the growing acyl chain, and *Mecr*, encoding the terminal reductase in each cycle of two-carbon unit addition (*Figure 1A*). It is important to note that because mtFAS is a cycle that requires all enzymes for each stepwise two-carbon addition, loss of each individual enzyme blocks fatty acid synthesis at an early, albeit distinct step. *Mcat* mutants should be unable to attach any carbon to ACP, whereas *Oxsm* mutants should be able to produce malonyl-ACP, and *Mecr* mutants could at most build short four-carbon acyl chains. We observed roughly 70% editing efficiency for all six of the guides utilized (based on T7E1 assays, *Figure 1—figure supplement 1A*). We found that transfection with Cas9 and sgRNAs targeting mtFAS genes led to smaller colonies at one-week post-single cell sorting relative to control

guides (*Figure 1—figure supplement 1B*). Many of the smallest clones stopped growing and/or did not survive expansion. After screening more than 100 of the surviving single cell clones, we failed to identify a single null mutant for any of the three genes targeted. This finding was not altogether unexpected, given that at least one other knockdown study concluded that NDUFAB1, the mammalian mitochondrial ACP, is essential in HEK293T cells (*Feng et al., 2009*). Indeed, The Broad Institute's Depmap lists NDUFAB1 as a 'common essential' gene, and the mtFAS genes all display negative gene effects, indicating they are essential in at least some cell lines (*Broad, 2020*). Taken together, our results strongly support the conclusion that the mtFAS pathway is essential in C2C12 skeletal myoblasts.

Although no complete null clones were generated, we were able to isolate several clonal cell lines with markedly decreased abundance of MCAT, OXSM, and MECR. These clones also had no detectable lipoylation of PDH and OGDH subunits (DLAT and DLST, respectively) in isolated mitochondrial fractions (*Figure 1B*), and grew slowly relative to control clones (*Figure 1—figure supplement 1C*). Growth of mtFAS mutant cells in both glucose and galactose, the latter of which requires mitochondrial respiration, was normalized by re-expression of the cognate mtFAS gene (*Figure 1C*, *Figure 1—figure supplement 1D and E*). MECR has been reported to have dual localization to the cytoplasm in addition to mitochondria (*Kim et al., 2014*), so we performed sub-cellular fractionation to assess these two MECR populations in our mutant cells. We observed an overall decrease in MECR expression that was most pronounced in the mitochondrial compartment (*Figure 1—figure supplement 1F*). We thus moved forward with the characterization of these presumably hypomorphic cell lines.

As mentioned previously, the canonical function of mtFAS is the production of octanoate, the eight-carbon precursor for lipoic acid synthesis. However, some have speculated that mtFAS might also generate fatty acids that contribute to phospholipid synthesis. Seminal studies found 3-hydroxymyristate as the predominant acyl modification on mitochondrial ACP in fungi (*Mikolajczyk and Brody, 1990*; *Schneider et al., 1997*). A more recent effort to identify the predominant lipid species found a variety of medium and long chain acyl species, raising the possibility of multiple products in addition to octanoate (*Angerer et al., 2017*). Although mtFAS mutant yeast display changes in steady-state lipids, the interpretation of this observation is confounded by other phenotypes of these cells (*Schneider et al., 1995*). In the only study that examines the role of mtFAS in lipid synthesis in mammals, transient knockdown of ACP did not change the abundance of mitochondrial lipids; however, this does not exclude the possibility that cytoplasmic FAS can compensate for loss of mtFAS (*Clay et al., 2016*).

We therefore set out to test whether mtFAS contributes to cellular phospholipid pools in the presence and absence of the cytosolic fatty acid synthase, FASN. In wild type and *Oxsm* mutant cell lines, we stably expressed hairpins targeting *FASN*, which encodes the cytoplasmic fatty acid synthase, or a scrambled control (*Figure 1D*). We observed no effect of *FASN* knockdown on protein lipoylation (*Figure 1D*). We then fed the cells uniformly labeled $^{13}$C-glucose and observed labeling in cellular lipids to define the relative contributions of cytoplasmic FAS and mtFAS to various cellular lipid pools. By monitoring accumulation of the m+two isotopologue over time, we observed a dependence on FASN expression, but not mtFAS for the synthesis of new phospholipids (shown are two representative PC species, *Figure 1E and F*). Higher order isotopologues from additional cycles of elongation by fatty acid synthesis (e.g. m + 4, m + 6, m + 8, and so on) did not show labeling above background in this experiment. Similarly, fatty acid methyl ester analysis (FAMES) showed an effect of *FASN* knockdown, but not *Oxsm* mutation, on incorporation of U$^{13}$C-glucose into c16:0 and c18:0 lipids, which confirmed that the m+2 labeling of phospholipids is likely in the fatty acid tails (*Figure 1—figure supplement 1G*). These experiments demonstrate that longer acyl chains synthesized by mtFAS do not contribute to cellular fatty acid and phospholipid pools and likely perform some other function.

## Mutation of mtFAS results in a profound mitochondrial respiratory phenotype and loss of ETC complexes

Given the known and hypothesized functions of mtFAS, we next assayed mitochondrial function in the mtFAS mutant cells. We performed a standard Seahorse mitochondrial stress test in high glucose medium and found that the mtFAS mutant cells displayed a roughly 30–40% decrease in basal respiration rate compared with controls (*Figure 2A*). More strikingly, the mtFAS mutants exhibited a lack of spare respiratory capacity, with FCCP-stimulated uncoupled respiration rates that were similar to

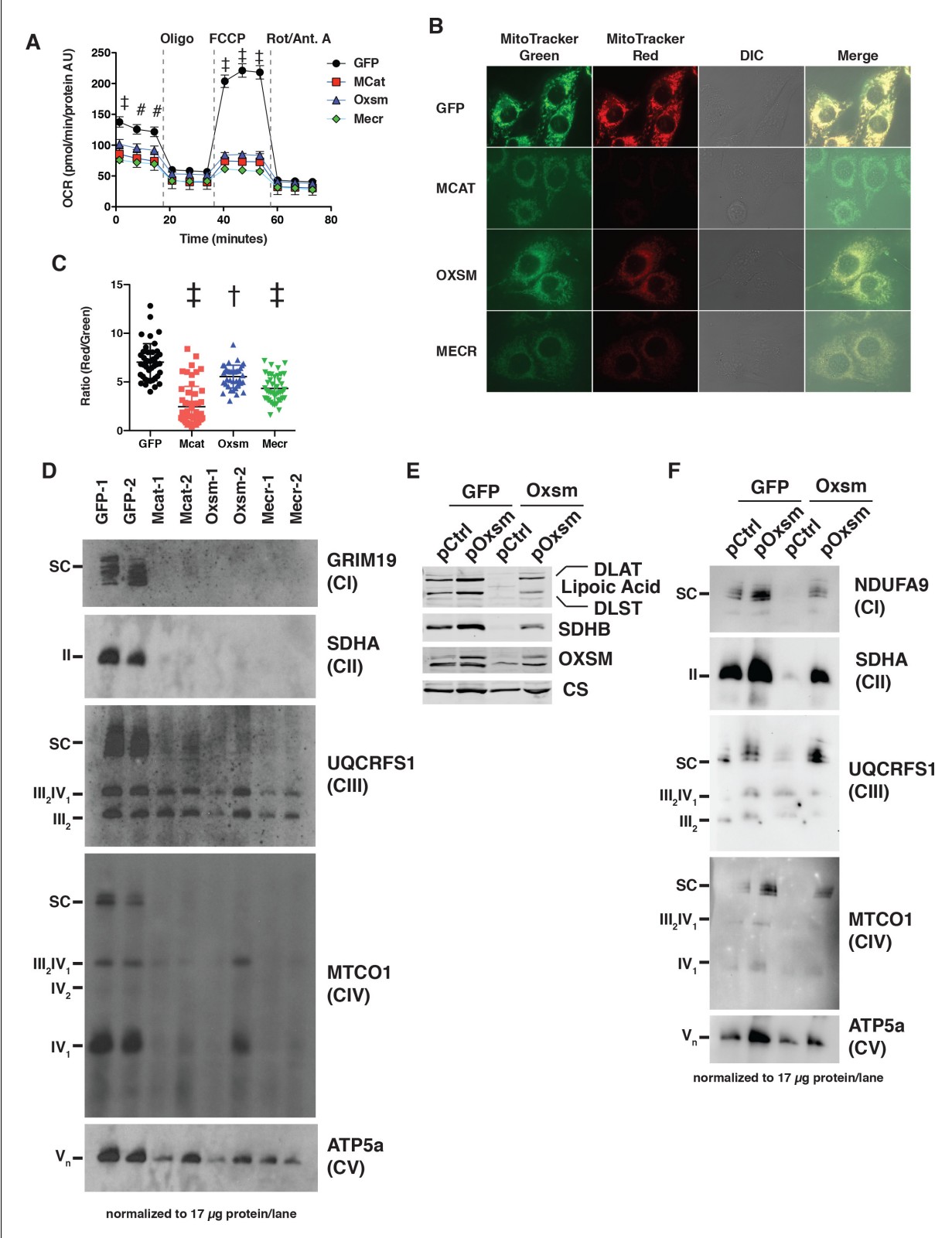

**Figure 2.** MtFAS mutants display profound loss of mitochondrial respiration and ETC complexes. (**A**) Cells from three clones of each of the indicated genotypes were seeded in eight wells of a 96-well seahorse plate and allowed to adhere overnight, then equilibrated and treated with the indicated drugs following standard mitochondrial stress test protocols from the manufacturer to determine Oxygen Consumption Rate (OCR). #=p < 0.01, †=p < 0.001, ‡=p < 0.0001 all comparisons are to GFP control, error bars are SEM. (**B**) Cells of the indicated genotype were seeded in chambered

*Figure 2 continued on next page*

*Figure 2 continued*

coverglass slides, stained with Mitotracker Red and Mitotracker Green, and imaged. (**C**) Ratio of MitoTracker Red to MitoTracker Green fluorescence of cells from 30 fields of view quantified using Fiji ImageJ. *=p < 0.05, †=p < 0.001, ‡=p < 0.0001 all comparisons are to GFP control, error bars are SD. (**D**) Mitochondrial lysates generated from the indicated cell lines were normalized for total protein by BCA assay, incubated with 1% digitonin, then separated by blue-native PAGE and immunoblotted with the indicated antibodies.

The online version of this article includes the following figure supplement(s) for figure 2:

**Figure supplement 1.** ETC complex assembly and activity is regulated by interaction of LYRM proteins with ACP.

the initially measured basal respiration (*Figure 2A*). Interestingly, *Oxsm* and *Mecr* mutants exhibited an increased basal extracellular acidification rate, suggesting that these cell lines compensate for decreased respiration via increased glycolysis, whereas *Mcat* mutants do not (*Figure 2—figure supplement 1A*). In agreement with the observed decrease in cellular respiration, mtFAS mutants also show a markedly decreased mitochondrial membrane potential, as measured by the ratio of Mito-Tracker Red to MitoTracker Green fluorescence (*Figure 2B,C* and *Figure 2—figure supplement 1B*).

Concurrent decreases in mitochondrial membrane potential and respiration can have several causes ranging from altered substrate utilization and TCA cycle activity to decreased expression and/or activity of electron transport chain (ETC) complexes. Therefore, we examined OXPHOS complex expression and assembly via blue-native PAGE analysis. Strikingly, fully assembled ETC complex I (CI), complex II (CII), and complex IV (CIV) were almost completely absent in mtFAS mutant mitochondria (*Figure 2D*). Complex V was also decreased in abundance, albeit to a lesser magnitude than that seen for CI, CII, and CIV. In contrast, complex III (CIII) was relatively unaffected by loss of mtFAS, although CIII-containing supercomplexes (SC) were absent, likely resulting from loss of CI and CIV. Furthermore, while CIII abundance and assembly were relatively unaffected, its activity was reduced in each of the mtFAS-deficient cells (*Figure 2—figure supplement 1C*). Altogether, these results depict a severe respiratory deficiency in mtFAS mutant cells, characterized by the striking loss of stably assembled OXPHOS complexes in mtFAS mutant mitochondria. Notably, this contrasts sharply with FASN, deletion of which does not affect fatty acid oxidation in muscle (*Funai et al., 2013*).

## mtFAS supports ETC assembly via the post-translational stabilization of LYRM proteins

To further examine the mechanisms underlying loss of ETC complexes in mtFAS mutant cells, we performed an unbiased quantitative proteomics experiment on duplicate whole cell lysates from three mtFAS mutant clones (one *Oxsm* and two *Mecr* mutants) and two control clones. Overall, there was no general trend for the steady-state abundance of mitochondrial proteins, with 77 proteins being statistically more abundant and 34 proteins being decreased in abundance (*Figure 3A*). We also observed that the majority of the proteins that comprise OXPHOS complexes were likewise similar in abundance between control and mtFAS mutant cells, including those in Complexes I, II and IV (*Figure 3A,B*), despite the decrease in abundance of the completely assembled complexes by blue-native PAGE (*Figure 2D*). However, a subset of eight OXPHOS proteins displayed significantly decreased abundance in both *Oxsm* and *Mecr* mutant cell lines compared to controls, including NDUFA6, NDUFA12, NDUFS4, NDUFS6, SDHB, COX5a, COX5b, and ATPIF1 (*Figure 3A,B*). To test whether these changes were transcriptional, we performed RNAseq analysis on control and mtFAS mutant cells. However, we found no significant changes in the abundance of transcripts encoding ETC component proteins (*Figure 3C* and *Figure 3—figure supplement 1A–C*), implying that the observed decrease in abundance of these proteins is likely due to post-translational regulation.

The two proteins that were most decreased in abundance, NDUFA6 and SDHB (*Figure 3D,E*), also stood out because of their relationships with the leucine-tyrosine-arginine motif (LYRM) protein family, a family of small proteins chiefly comprised of late-stage ETC assembly factors (*Angerer, 2015*). In many cases, such as for Complex II, LYRM proteins (SDHAF1 and SDHAF3) bind to and facilitate the insertion of a target protein (in this case SDHB) in the final stage of ETC complex assembly (*Ghezzi et al., 2009*; *Na et al., 2014*). In other cases, such as for Complex I, the LYRM proteins NDUFA6 and NDUFB9 are actually stable subunits of the fully assembled complex

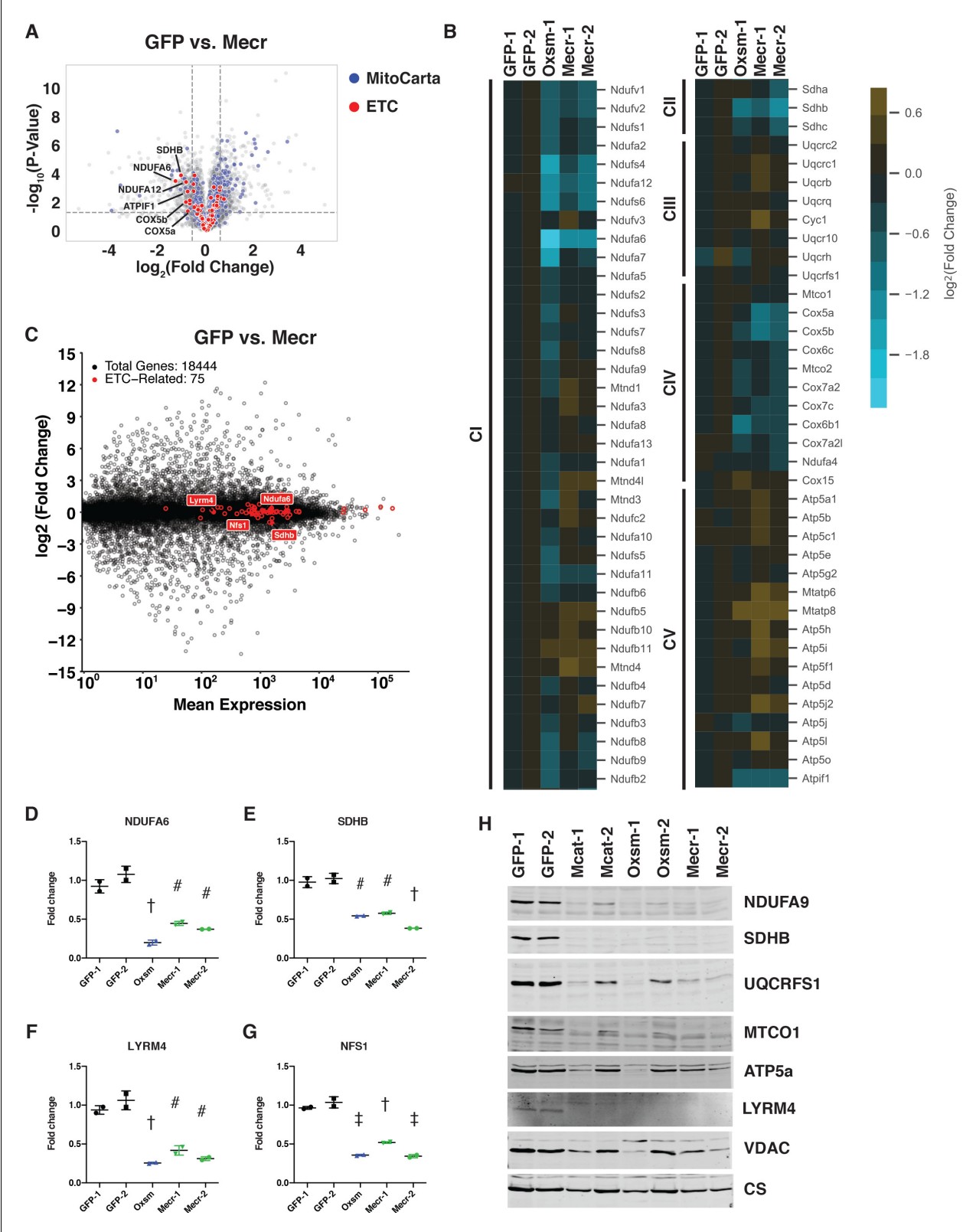

**Figure 3.** Posttranslational loss of ETC components in mtFAS mutants is specific to LYR proteins and their targets. (**A-B**) Duplicate samples from the indicated cell lines were grown under proliferative conditions and subjected to TMT labeling and quantitative proteomics analysis. (**A**) Volcano plot of compiled Mecr clones vs. GFP controls showing all proteins (gray), mitochondrial proteins (blue), and electron transport chain subunits (ETC, red). Dashed gray lines indicate cutoffs for significance at -log10(p-value) = 1.3 and log2(Fold Change) = +/- 0.59. (**B**) Heatmap depicting log2(Fold Change)

*Figure 3 continued on next page*

*Figure 3 continued*

of OXPHOS subunits in the indicated cell lines. (C) Quadruplicate samples from mtFAS mutant cells and controls were grown under proliferative conditions. Total RNA was isolated, used as input for mRNA library prep, and sequenced. Resulting data were aligned to the mouse genome and analyzed for differential expression. ETC subunit-encoding transcripts are shown in red vs. all other transcripts (gray). (D-G) Relative abundance of the indicated LYR proteins or their targets in the indicated cell lines from the quantitative proteomics experiment described in (A-B) #=p < 0.01, †=p < 0.001, ‡=p < 0.0001, error bars are SD. All statistical comparisons shown are between mtFAS mutants and GFP-1 clone; p-values when compared with GFP-2 clone were similar or smaller than when compared with GFP-1 clone. (H) Crude mitochondrial lysates generated from the indicated cell lines by differential centrifugation were normalized for total protein by BCA assay, separated by SDS-PAGE, and immunoblotted with the indicated antibodies.

The online version of this article includes the following figure supplement(s) for figure 3:

**Figure supplement 1.** Transcription of ETC subunit encoding genes and translation of mitochondrially encoded proteins are unaffected in mtFAS mutants.

(*Fiedorczuk et al., 2016*; *Zhu et al., 2016*). LYRM proteins are also found as a stable component of the NFS-containing iron-sulfur cluster biogenesis (ISC) complex (LYRM4) as well as in an assembly intermediate of the mitochondrial ribosome (AltMid51) (*Boniecki et al., 2017*; *Brown et al., 2017*). High-throughput interactomics studies in mammalian systems and targeted experiments in yeast have identified physical interactions between ACP and several LYRM proteins (*Floyd et al., 2016*; *Huttlin et al., 2015*; *Majmudar et al., 2019*; *Van Vranken et al., 2018*). Interestingly, the ACP protein found in complex with LYRM proteins appears often to have maintained its acyl modification, which is generated through the sequential actions of the mtFAS enzymes as described above (*Angerer et al., 2014*; *Cory et al., 2017*; *Runswick et al., 1991*; *Van Vranken et al., 2016*). Studies have shown that the acyl chain on ACP is intimately involved in these physical interactions, folding into the middle of the LYRM proteins, but the role of ACP acylation in its varied functions is mostly untested (*Angerer et al., 2017*; *Boniecki et al., 2017*; *Cory et al., 2017*; *Fiedorczuk et al., 2016*; *Zhu et al., 2016*).

In addition to the structurally verified ACP-LYRM interactions, three other LYRM family members with known target proteins have been suggested to interact with ACP based on high-throughput interactomics studies: SDHAF1 and SDHAF3, discussed above, and LYRM7, which mediates the addition of UQCRFS1 in the last step of complex III assembly (*Floyd et al., 2016*; *Huttlin et al., 2015*; *Na et al., 2014*; *Sánchez et al., 2013*). Thus, we hypothesized that the decreased abundance of SDHB (*Figure 3E*) could be explained by interaction of ACP with SDHAF1 and/or SDHAF3. To verify whether ACP in fact interacts with these additional LYRM family members, we expressed epitope-tagged variants of ACP and each LYRM in cells and confirmed their physical interaction via co-immunoprecipitation and immunoblotting (*Figure 3—figure supplement 1D*). Along with other published data (*Van Vranken et al., 2018*), the finding that NDUFA6, a known LYRM protein, and SDHB, the target of two known LYRM proteins, exhibit decreased abundance in mtFAS mutant cell lines supports the hypothesis that interaction with an acylated ACP is required for the complex assembly functions of these LYRM proteins.

This finding prompted us to examine the abundance of other LYRM proteins and their targets in the mtFAS mutant cell lines. Indeed, LYRM4 and its target NFS1 were also significantly decreased in abundance in our whole cell proteomics dataset, which was confirmed by western blot, implying that LYRM4 also requires acylated ACP for the stability and function of itself and NFS1 (*Figure 3F–H*). Interestingly, NDUFB9, the other CI LYRM, and UQCRFS1, a subunit of CIII and the target of LYRM7, were not significantly decreased in abundance in mtFAS mutant cells (*Figure 3—figure supplement 1E,F*), implying that either the interactions between ACP and these LYRM proteins is not dependent on acylation and mtFAS, or that other compensatory mechanisms are at play. We performed immunoblot analysis of UQCRFS1 and found that it was also decreased in abundance similarly to other LYR targets, perhaps explaining the observed decrease in CIII activity despite the partial maintenance of complex assembly (*Figure 3H*).

Finally, ACP has also been found to interact with an assembly intermediate of the mitochondrial ribosome through a novel LYRM protein called altMiD51 and its target, MALSU1 (*Brown et al., 2017*). Brown et al. proposed that this interaction might negatively regulate mitochondrial translation because MALSU1, altMiD51, and ACP bind the large mitochondrial ribosomal subunit in a way that precludes small subunit binding, but the role of ACP acylation in this interaction is unclear

(*Brown et al., 2017*; *Dibley et al., 2020*; *Rathore et al., 2018*). Despite this proposed role of ACP in mitochondrial translation, we found that the abundance of mitochondrially encoded proteins was unchanged in the mtFAS mutant cell lines (*Figure 3—figure supplement 1G*). Importantly, this includes MTCO1 and MTCO2, two subunits of CIV, strongly implying that the diminution of CIV in mutant cells does not result from loss of mitochondrial translation. Interestingly, however, MALSU1 was significantly decreased in abundance (*Figure 3—figure supplement 1H*), which raises the possibility that MALSU1 downregulation might be a compensatory adaptation that the cells make to avoid collapse of mitochondrial translation upon loss of mtFAS.

## Impairment of mtFAS induces reductive labeling of TCA cycle intermediates

The profound defects observed in mitochondrial respiration led us to examine mitochondrial and cellular metabolism more comprehensively in our mtFAS mutants. We first performed steady-state metabolomics analysis on control and mtFAS mutant cells. Purines were among the most significantly depleted metabolites in mtFAS mutants, but we observed no change in pyrimidines (*Figure 4—figure supplement 1A,B*, all measured metabolites in *Supplementary file 1*). Purine depletion may be a result of decreased activity of another lipoylated protein – the glycine cleavage H protein, which plays a role in purine biosynthesis (*Fujiwara et al., 1991*). mtFAS mutants exhibited robust pyruvate accumulation, in agreement with the decreased lipoylation and reduced activity of PDH, but only a minor and statistically insignificant increase in intracellular lactate (*Figure 4A* and *Figure 4—figure supplement 1C*). Similarly, no significant changes were observed in the abundance of glutamine or glutamate (*Figure 4—figure supplement 1D,E*). In contrast, the TCA cycle intermediates citrate, fumarate, and malate all showed significantly decreased abundance (*Figure 4B–D*). Aspartate, the synthesis of which requires the TCA cycle metabolite oxaloacetate and an electron acceptor such as the ETC (*Birsoy et al., 2015*; *Sullivan et al., 2015*), was strongly depleted in the mtFAS mutant cells (*Figure 4E*).

To further understand how TCA cycle metabolism is affected by impairment of mtFAS, we fed the cells uniformly labelled $^{13}$C-glutamine for 24 hr and assessed labeling of TCA cycle intermediates to monitor glutamine contribution to TCA cycle flux (schematic in *Figure 4F*). This experiment led to three observations that were particularly interesting. First, glutamate cycling in the TCA cycle is decreased in the mtFAS mutant cells. We observed no differences in glutamine uptake in mutant cells compared to controls (*Figure 4—figure supplement 1F*). However, in pre-steady state measurements we observed reduced labeling of glutamate in mtFAS mutants (*Figure 4—figure supplement 1G*). Furthermore, after 24 hr of labeling, the mtFAS-deficient cells showed reduced fractional enrichments of the glutamate m+three and m+one isotopologues, which are synthesized on subsequent turns of the TCA cycle (*Figure 4G*). In contrast, the m+five isotopologue contributes a larger fraction of the total pool (*Figure 4G*). These data are consistent with decreased processivity of the TCA cycle in cells lacking mtFAS.

Secondly, we observed that labeling of TCA cycle intermediates shows a clear distinction between succinate and later metabolites. Like glutamate, succinate labeling is initially slow (*Figure 4—figure supplement 1H*). After 24 hr in $^{13}$C-glutamine, however, similar to what we observe for glutamate labeling, in control cells subsequent rounds of the TCA cycle cause m+two succinate to accumulate. In mtFAS mutant cells, m+two succinate labeling is reduced while m+four accumulates, indicating its decreased cycling in the TCA cycle (*Figure 4H*). In contrast, m+4 labeling of fumarate, malate, and citrate are all markedly lower at short time points (*Figure 4—figure supplement 1I–K*), and remain low after 24 hr (*Figure 4I–K*), along with m+four aspartate, which is derived from m+four citrate (*Figure 4L*). These data and the reduced abundances of fumarate and malate support a block of the TCA cycle at the succinate dehydrogenase (SDH) catalyzed step, and is in agreement with our data that demonstrate decreased steady state SDHB stability and a near-complete loss of fully assembled Complex II (*Figures 2D* and *3H*).

Finally, the third observation we made from these data is that mtFAS mutants displayed significantly increased levels of m+five citrate (*Figure 4K*), which is formed from the reductive carboxylation of α-ketoglutarate, and is the canonical marker of reductive carboxylation. Accordingly, we also saw significant increases in m+three fumarate, malate, and aspartate (*Figure 4I,J and L*), which are formed from this pathway downstream of m+five citrate (*Mullen et al., 2011*). Previous analysis of the reductive pathway concluded, as we observed here, that cells with ETC defects continue to

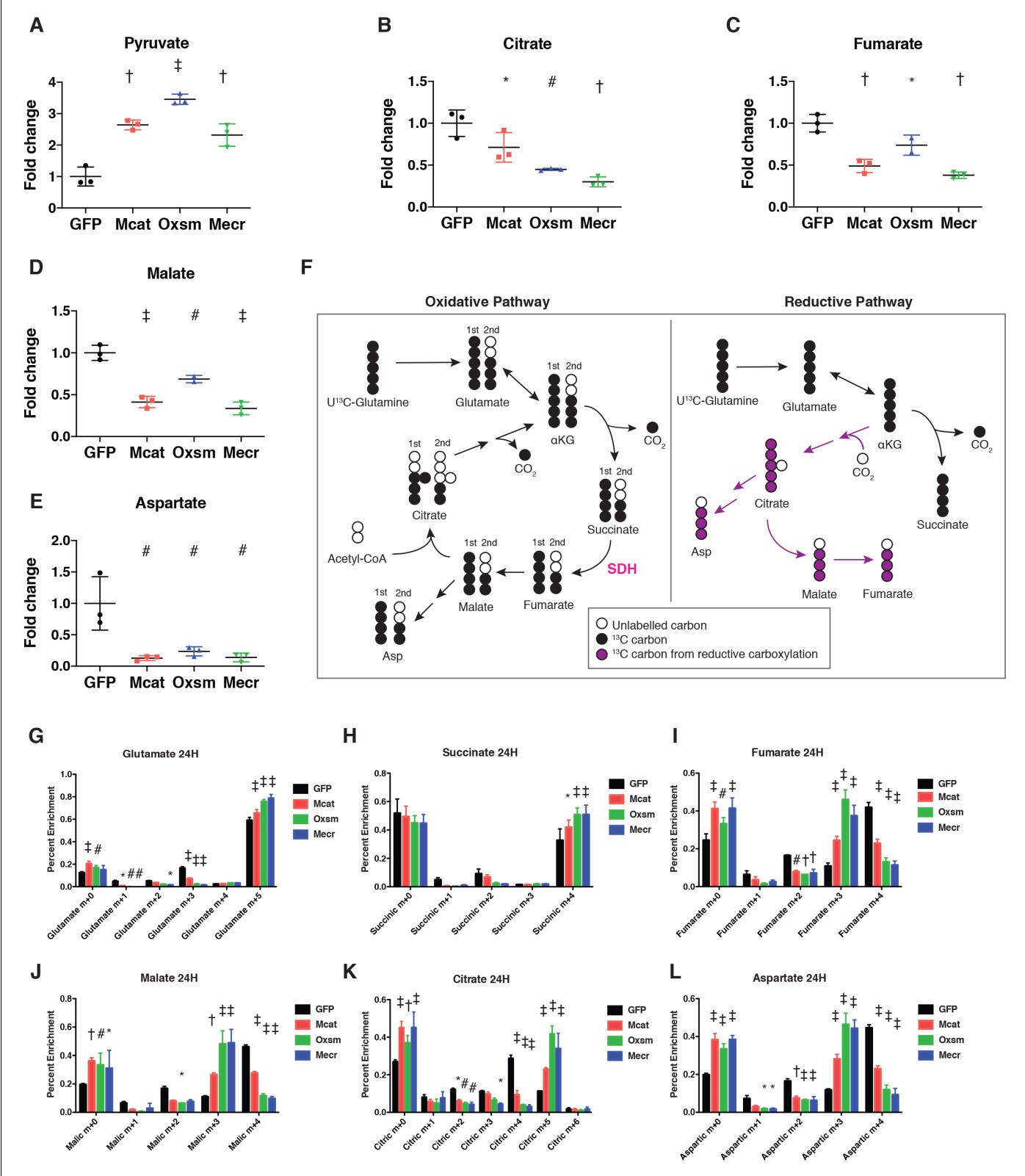

**Figure 4.** Impairment of mtFAS promotes switch from oxidative to reductive mitochondrial metabolism. (A–E) Triplicate biological samples from mtFAS mutant cell lines or GFP control were grown under standard proliferative conditions in high glucose medium (25 mM) and harvested for steady-state metabolomics analysis by LC-MS. Shown are relative pool sizes for the indicated metabolites. *=p < 0.05, #=p < 0.01, †=p < 0.001, ‡=p < 0.0001, error bars are SD. (F) Schematic of isotopomeric labeling of TCA cycle intermediates upon feeding with U$^{13}$C-glutamine. Black circles indicate $^{13}$C carbons

*Figure 4 continued on next page*

*Figure 4 continued*

derived from labeled glutamine via oxidative TCA cycle flux (left). Purple circles indicate $^{13}$C carbons derived from labeled glutamine via reductive carboxylation (right). (G-L) Triplicate biological samples of the indicated genotype were labeled for 24 hr with U$^{13}$C-glutamine, harvested, and analyzed via GC-MS for the indicated metabolites and their isotopologues. *=p < 0.05, #=p < 0.01, †=p < 0.001, ‡=p < 0.0001, error bars are SD.

The online version of this article includes the following figure supplement(s) for figure 4:

**Figure supplement 1.** Impairment of mtFAS causes changes in purine abundance and TCA cycle flux.

produce succinate by the oxidative pathway, perhaps to generate reducing equivalents to drive reductive carboxylation (*Mullen et al., 2014*). In summary, these data support the conclusion that oxidative cycling of TCA cycle intermediates is decreased overall, with a particular block at the SDH-catalyzed step. Furthermore, mtFAS mutants invoke reductive carboxylation to produce citrate, and reductive labeling of downstream metabolites is also increased. Reductive carboxylation is often seen in models of ETC inhibition or mutation (*Mullen et al., 2011*) and further supports the model that hypomorphic mtFAS mutant skeletal myoblasts display significant ETC dysfunction.

## Loss of ETC assembly and reductive carboxylation are not downstream of loss of protein lipoylation

The best-studied function of mtFAS is the biosynthesis of octanoate, the eight-carbon precursor of lipoic acid, but it also produces longer acyl chains. This raises the possibility that there are at least two independent mechanisms whereby mtFAS might act on mitochondrial function: one via lipoic acid synthesis and another in which these longer acyl chains facilitate the acyl-ACP dependent activation of LYRM proteins in ETC complex assembly. Therefore, we decided to determine whether the metabolic phenotypes observed in mtFAS mutant cells are also found in cells containing an intact mtFAS system, but lacking the ability to transfer lipoic acid to target proteins, including the 2-ketoacid dehydrogenases (2KADHs). We created mutant cell lines for *Lipt1*, the gene that encodes the terminal enzyme in protein lipoylation (*Figure 5A*). In these mutants, synthesis of octanoate and longer acyl chains by mtFAS should be unaffected, as Lipt1 catalyzes the terminal step in lipoic acid transfer to its target proteins downstream of mtFAS (reviewed in *Solmonson and DeBerardinis, 2018*). Similar to mtFAS mutants, *Lipt1* mutant cells displayed undetectable protein lipoylation (*Figure 5A*), with no change in the expression of MCAT, OXSM, and MECR proteins (*Figure 5—figure supplement 1A*). The fact that mtFAS mutants and *Lipt1* mutants both have drastically reduced protein lipoylation, below the limit of detection via western blotting, allows us to attribute any phenotypes observed in mtFAS mutants that are not phenocopied by *Lipt1* mutants to non-lipoic acid products of mtFAS.

Like mtFAS mutants, Lipt1 mutants display an analogous, albeit somewhat milder respiratory phenotype (*Figure 5B*). However, in sharp contrast to the destabilization of ETC complexes observed in mtFAS mutants, *Lipt1* mutants largely maintain an assembled ETC, with similar levels of CI, CII, and SC as in control cells (*Figure 5C*). Notably, CV was more destabilized in *Lipt1* mutants than in mtFAS mutants, suggesting a mtFAS-independent mode of CV regulation by LIPT1 outside of lipoic acid (*Figure 5C*). Nevertheless, it appears that the mtFAS-dependent regulation of CI and CII is essentially independent of lipoic acid synthesis and therefore likely involves some other lipid product of mtFAS.

Finally, we wanted to determine which portions of the metabolic phenotypes resulting from mtFAS loss are attributable to loss of protein lipoylation. We found that, in contrast to mtFAS mutants, *Lipt1* mutants had normal abundance of TCA cycle metabolites (*Figure 5D*). U$^{13}$C-glutamine flux experiments showed that LIPT1 loss caused a reduction in oxidative labeling and at most a small increase in reductive labeling of fumarate and malate (*Figure 5E*), consistent with its small effects on OCR and ETC assembly and with similar labeling effects in fibroblasts from LIPT1-deficient patients (*Ni et al., 2019*). In all, our data strongly support a model in which mtFAS-dependent acyl chains facilitate acyl-ACP/LYRM interactions to support the biosynthesis of ETC complexes independent of lipoic acid.

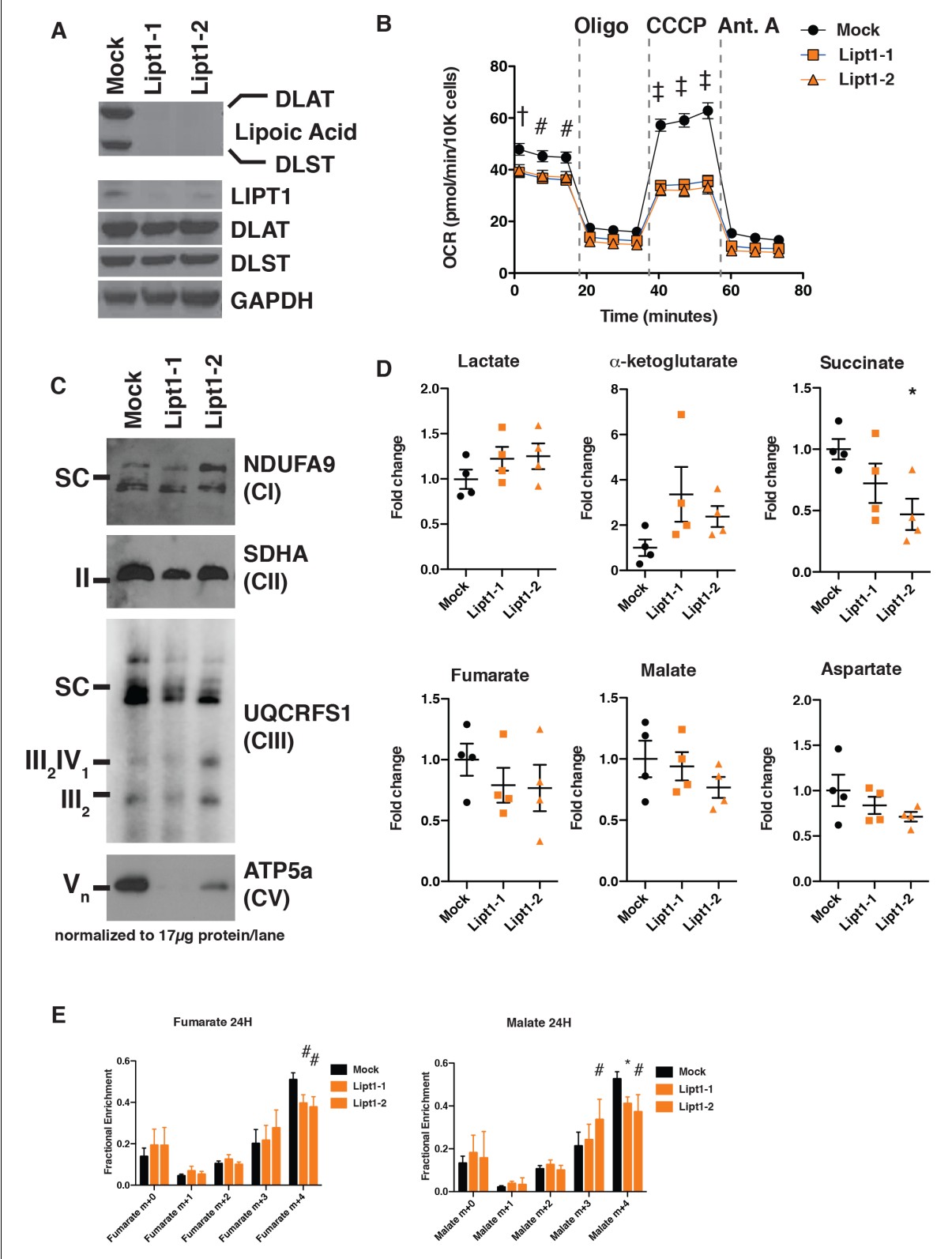

**Figure 5.** MtFAS mutant phenotypes are not recapitulated by loss of lipoic acid alone. (A) Immunoblot for the indicated proteins in whole cell lysates from Lipt1 mutant and control cell lines. (B) In triplicate experiments, cells of each of the indicated clones were seeded in eight wells of a 96-well seahorse plate and allowed to adhere overnight, then equilibrated and treated with the indicated drugs following standard mitochondrial stress test protocols from the manufacturer to determine Oxygen Consumption Rate (OCR). #=p < 0.01, †=p < 0.001, ‡=p < 0.0001 all comparisons are to GFP

*Figure 5 continued on next page*

*Figure 5 continued*

control, error bars are SEM. (**C**) Mitochondrial lysates generated from the indicated cell lines by differential centrifugation were normalized for total protein by BCA assay, incubated with 1% digitonin, then separated by BN-PAGE and immunoblotted with the indicated antibodies. (**D**) Four biological replicates from mtFAS mutant cell lines or GFP control were grown under standard proliferative conditions and harvested for steady-state metabolomics analysis by LC-MS. Shown are relative pool sizes for the indicated metabolites. *=p < 0.05, error bars are SD. (**E**) Four biological samples of the indicated genotype were labeled for 24 hr with U$^{13}$Cglutamine, harvested, and analyzed via GC-MS for the indicated metabolites and their isotopologues. *=p < 0.05, #=p < 0.01, †=p < 0.001, error bars are SD.

The online version of this article includes the following figure supplement(s) for figure 5:

**Figure supplement 1.** Expression of mtFAS proteins is unchanged in *Lipt1* mutant cell lines.

## Impairment of mtFAS impedes skeletal myoblast differentiation

C2C12 myoblasts are muscle precursor cells that remain proliferative in culture unless stimulated to terminally differentiate. However, upon examination of the quantitative proteomics experiment described above, we noticed that the largest group of proteins with decreased abundance were all related to the terminal differentiation of muscle cells (*Figure 6A*, *Supplementary file 2*). Although some of these differences did not reach statistical significance, likely due to the low abundance of these proteins in proliferative myoblast cultures, we were nonetheless intrigued by this observation, which might imply a defect in myoblast differentiation. Indeed, when treated with differentiation medium for up to seven days, mtFAS mutant cell lines almost completely failed to differentiate, as monitored by the formation of multinucleated myotubes (*Figure 6B* and *Figure 6—figure supplement 1A*).

Muscle cell differentiation is controlled by a transcription factor cascade in which MyoD activates the transcription of Myogenin (Myog), and Myog in turn activates the transcription of many terminal differentiation genes, such as myosin heavy chain (MHC, encoded by *Myh1*) (*Zammit, 2017*). To further explore the effect of mtFAS mutation on differentiation, we immunoblotted for MyoD, Myog, and MHC in proliferative conditions (day −1) and at 1, 3, and 7 days of culture in differentiation medium. Although MyoD expression was largely unaffected in mtFAS mutant cells, Myog induction appeared to be decreased or absent in mtFAS cells (*Figure 6C*). Similarly, the Myog target protein MHC was almost undetectable in mtFAS mutant cells (*Figure 6C*). To further explore the mechanism underlying impairment of Myog induction and muscle cell differentiation, we performed RNA-Seq analyses on cells grown at early time points in differentiation (day −1, day 0, and day 1). The results confirmed no difference in *MyoD* transcript expression, but showed both a delay and blunting of induction of the *Myog* transcript as well as those of its target genes such as *Myh1* and myogenic differentiation genes generally (*Figure 6D,E* and *Figure 6—figure supplement 1B–D*). Another Myog target, the transcription factor Mef2c, is also involved in the feed forward propagation of the myogenic program (*Ji et al., 2009*; *Rogerson et al., 2002*). Interestingly, in two of the three mtFAS mutant clones assayed, *Mef2c* transcripts were also strikingly down-regulated compared with control (*Figure 6D,E*), which raises the possibility that it plays a particularly important role in the connection of mtFAS and mitochondrial metabolism with myocyte differentiation.

*Myog* induction and myogenic differentiation requires extensive epigenetic remodeling (*Asp et al., 2011*). Several cellular demethylases, including DNA demethylases and Jumonji histone demethylases use alpha-ketoglutarate (αKG) as a methyl acceptor, and the αKG to succinate ratio has been shown to affect demethylase activity (*Pavlova and Thompson, 2016*; *Soloveychik et al., 2016*). Steady state metabolomics analysis of mtFAS mutant cell lines revealed a large increase in αKG as well as in the αKG/succinate ratio (*Figure 6F*), suggesting a possible mechanism for the impairment of *Myog* induction and myocyte differentiation.

## Discussion

Here we have shown that three genes in the mtFAS pathway (*Mcat*, *Oxsm*, and *Mecr*) are likely essential in cultured mammalian skeletal myoblasts. Hypomorphic mutants of these genes lead to loss of detectable protein lipoylation, without affecting structural phospholipids such as PC. Strikingly, mtFAS impairment results in a profound mitochondrial phenotype characterized by loss of assembled ETC complexes I, II, and IV. Quantitative proteomics analysis revealed the specific

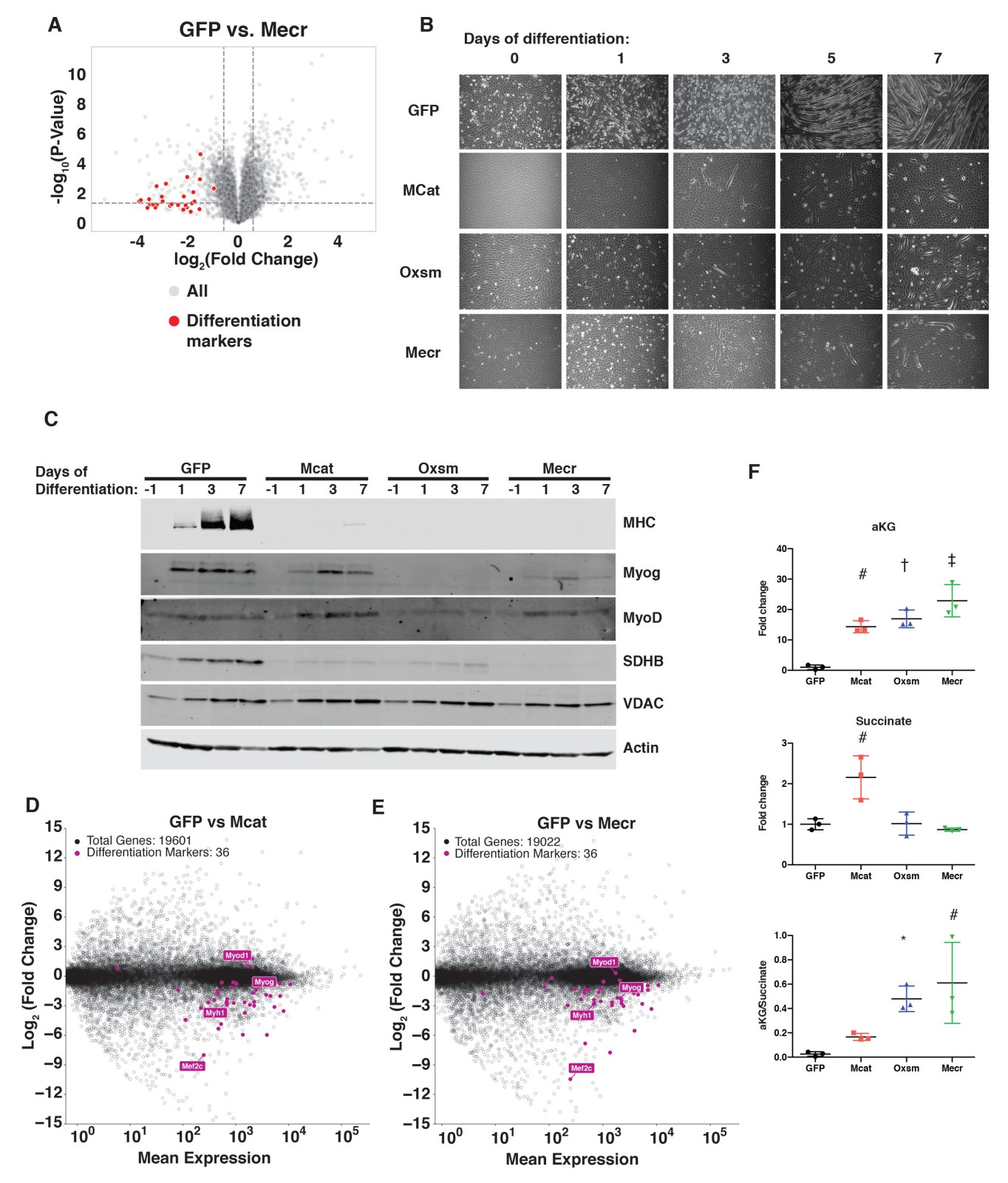

**Figure 6.** MtFAS is required for skeletal myoblast differentiation. (**A**) Volcano plot of quantitative proteomics experiment showing Mecr mutant samples (n = 4) vs. GFP (n = 4) controls; all proteins (gray), and proteins associated with the differentiated skeletal muscle lineage (red). Dashed lines indicate cutoffs for significance at -$\log_{10}$(p-value) = 1.3 and $\log_2$(Fold Change) = +/- 0.59. (**B-C**) Cells of the indicated genotypes were plated and grown for 24 hr under proliferative conditions (day −1), then switched to 2% horse serum differentiation medium at 95% confluency (day 0). Medium was replaced daily

*Figure 6 continued on next page*

*Figure 6 continued*

for the indicated number of days. Plates were imaged (**B**) or whole cell lysates were collected at the indicated time points, separated by SDS-PAGE, and immunoblotted for the indicated proteins (**C**). (**D-E**) Quadruplicate samples from mtFAS mutant cells and controls were grown under proliferative conditions to 95% confluency then switched to 2% horse serum differentiation medium for 24 hr. Total RNA was isolated, used as input for mRNA library prep, and sequenced. Resulting data were aligned to the mouse genome and analyzed for differential expression. Transcripts for differentiation related proteins from panel (**A**) are shown in purple vs. all other transcripts (gray). (**F**) Relative steady state pool sizes of the indicated metabolites in mtFAS mutant cell lines vs. GFP controls grown in proliferative conditions from steady-state LC-MS metabolomics analysis. *=p < 0.05, #=p < 0.01, †=p < 0.001, ‡=p < 0.0001, error bars are SD.

The online version of this article includes the following figure supplement(s) for figure 6:

**Figure supplement 1.** Skeletal myoblast differentiation is delayed in mtFAS mutant cell lines.

decreases in the abundance of two LYRM proteins, NDUFA6, a subunit of CI, and LYRM4, a member of the ISC complex, along with its target protein, NFS1, and another LYRM target, SDHB. Importantly, the LYRM proteins that coordinate SDHB in CII assembly, SDHAF1 and SDHAF3, were not identified in our proteomics dataset, however we infer that they are likely also decreased in abundance in mtFAS mutant cells, giving rise to the observed effect on SDHB. RNA-Seq analyses confirmed that the decreased abundance of these proteins is not a result of transcriptional downregulation, but likely reflects post-translational destabilization. Our data together with that from multiple other eukaryotic model systems support a model in which this effect is mediated through the physical interaction of acyl-ACP with LYRM proteins, resulting in their acylation-dependent activation and functional assembly of ETC complexes (*Angerer et al., 2017*; *Van Vranken et al., 2018*).

In contrast to mtFAS mutants, *Lipt1* mutants can still synthesize acyl chains on ACP, but fail to transfer lipoate to its target proteins downstream of mtFAS. Thus, like mtFAS mutants, *Lipt1* mutants lose detectable lipoylation of PDH and OGDH, and have impaired mitochondrial respiration. However, the acylation-dependent activation of LYRM proteins and ETC complex assembly appears to require lipid products of mtFAS other than lipoic acid, as although *Lipt1* mutants lose protein lipoylation, they maintain mitochondrial ETC complexes I-IV. This finding is supported by studies in yeast that reached a similar conclusion based upon a suppressor mutation which can rescue OXPHOS in the absence of lipoic acid restoration (*Kursu et al., 2013*; *Van Vranken et al., 2018*). We also found that mtFAS mutants and *Lipt1* mutants both undergo significantly decreased oxidative TCA cycling, however only mtFAS mutants appear to display decreased abundance of TCA cycle intermediates and robustly employ reductive carboxylation. We infer that these metabolic effects are likely downstream of loss of ETC complex function, as has been previously reported in the literature (*Mullen et al., 2011*). As both mtFAS mutants and *Lipt1* mutants have undetectable lipoylated PDH and OGDH, but only mtFAS mutants have impaired synthesis of longer acyl chains on ACP, these data suggest that it is these longer acyl chains that are important for the interaction of acyl-ACP and activation of LYRM proteins. These findings also have implications for how patients with mutations in the lipoic acid pathway vs. the mtFAS pathway are treated.

It is important to note that some of the effects we observe on ETC assembly cannot be explained by the known interactions between LYRMs and acyl-ACP. For example, it is unclear how mtFAS impairment leads to the instability of Complex IV. One possibility is simply that CIV is less stable outside of its association with respiratory supercomplexes, which are almost completely absent in mtFAS mutant cells, likely as a result of Complex I loss. Alternatively, COX5A and COX5B are among the most affected ETC subunits by proteomics, and could be the targets of either a novel LYRM protein or another mtFAS-dependent mechanism. In mammals, the LYRM family includes at least eleven proteins but as this motif is degenerate and poorly defined, there are likely to be many that are still undiscovered. Of the currently annotated LYRM proteins, at least three (LYRM1, LYRM2, and LYRM9) have no known molecular target, although LYRM2 has recently been implicated in CI assembly in addition to NDUFA6 and NDUFB9 (*Dibley et al., 2020*). The extent to which these novel LYRM proteins require ACP acylation for function is also unknown.

Similarly, further study into the difference(s) between LYRMs that depend on ACP acylation and those that do not is warranted. As mentioned above, we observed decreases in abundance of two

LYRM proteins, NDUFA9 and LYRM4, and can infer the destabilization of SDHAF1 and SDHAF3 from the decreased abundance of SDHB. Conversely, we did not observe destabilization of NDUFB9 and UQCRFS1 upon mtFAS mutation, implying that NDUFB9 and LYRM7 are not dependent on ACP acylation for their stability and functions, although orthologs in other eukaryotes seemed to be (*Angerer et al., 2017*; *Van Vranken et al., 2018*). However, it is possible that this effect could result from residual mtFAS activity, as complete knockout of the mtFAS pathway proved impossible in this system. Furthermore, at least in the case of UQCRFS1 and LYRM7, the maintenance of CIII may have been clonally selected for as cells require CIII function for quinone oxidation and dihydroorotate dehydrogenase activity to make uridine (*Khutornenko et al., 2014*; *Khutornenko et al., 2010*). Therefore, cells that lack CIII are auxotrophic for orotate or uridine, and lack of these metabolites in the culture medium may have provided selective pressure for clones that employ some mechanism to escape the mtFAS-dependent regulation of CIII. In fact, it is tempting to speculate that assembly of CIII may be an essential function of mtFAS in mammalian cells.

Finally, the question of why cells have evolutionarily maintained a parallel pathway for lipid synthesis in mitochondria remains interesting, especially given recent research demonstrating the transfer of lipids between endoplasmic reticulum, mitochondria, and various other organelles, implying that the answer cannot simply be compartmentalization (*Yang et al., 2018*). Acetyl-CoA is the nearly universal fuel of mitochondrial oxidative metabolism, being the convergence point of the catabolism of carbohydrates, fatty acids, ketones and some amino acids. Acetyl-CoA is also the common substrate of mtFAS and the TCA cycle, where it is broken down to produce the reduced NADH and FADH$_2$ that fuel the electron transport chain. Thus, the interaction of acyl-ACP with the LYRM proteins to facilitate ETC assembly has high potential to be a regulatory mechanism whereby acetyl-CoA regulates its own catabolism (*Kastaniotis et al., 2004*; *Kursu et al., 2013*; *Van Vranken et al., 2018*). In other words, as acetyl-CoA is the substrate for mtFAS, it is required for ACP acylation, which in turn is required for interaction with and activation of LYRM proteins and ETC assembly. The assembled ETC catalyzes the oxidation of NADH and FADH$_2$, which drives the TCA cycle and consumption of acetyl-CoA. Thus, when acetyl-CoA is low, ACP acylation is diminished, ETC assembly is blunted, TCA cycling slows, and acetyl-CoA consumption falls. This hypothesis is supported by the observation that yeast with defects in mitochondrial pyruvate metabolism exhibit blunted ACP acylation and impaired ETC assembly (*Van Vranken et al., 2018*). This elegant regulatory cycle provides cells with a potential mechanism to monitor acetyl-CoA substrate availability and adjust ETC complex levels accordingly, avoiding situations where the ETC sits idle in the absense of substrate, and reducing the attendant deleterious consequences, such as the production of reactive oxygen species. Similarly, it could potentially allow cells to ramp up oxidative capacity in settings of excess mitochondrial acetyl-CoA.

Human mutations in genes encoding enzymes of the mtFAS pathway are rare, but continue to be discovered (*Gorukmez et al., 2019*; *Heimer et al., 2016*; *Li et al., 2020*). Intriguingly, patients with *Mecr* mutations exhibit muscle weakness and we show herein that mtFAS mutation interferes with the normal differentiation of cultured myoblasts (*Figure 6*). The implications of these studies for the broader population remain to be seen, however. For example, many age-related human diseases exhibit evidence of decreased mitochondrial respiration, including frequent defects in OXPHOS complex assembly, suggesting the interesting possibility that alterations in mtFAS might contribute to mitochondrial impairments in human disease. The role(s) of mtFAS in aging and more prevalent diseases that also involve mitochondrial pathologies such as diabetes, heart disease, and cancer are essentially unstudied. One recent publication suggested that lipoylation of PDH and expression of *Oxsm* may be decreased in some tissues in diabetic mice (*Gao et al., 2019*). This raises the intriguing prospect that augmenting mtFAS pathway function could 'boost' or rescue mitochondrial oxidative metabolism in settings where mitochondrial respiration is sub-optimal. Recent studies have shown enhancement of mitochondrial metabolism upon ACP overexpression, although the molecular mechanisms and dependence of these phenotypes on mtFAS are unclear (*Hou et al., 2019*; *Zhang et al., 2019*). It is therefore important to thoroughly examine how this pathway functions both in normal biology and in disease settings. Understanding the rate limiting steps in the cycle, its mode of regulation, and the mechanisms through which it can be manipulated could one day allow mtFAS to be harnessed therapeutically to build better mitochondria and treat metabolic disease.

# Materials and methods

## Key resources table

| Reagent type (species) or resource | Designation | Source or reference | Identifiers | Additional information |
|---|---|---|---|---|
| cell line (*M. musculus*) | C2C12 | ATCC | #CRL-1772, RRID:CVCL_0188 | |
| cell line (*H. sapiens*) | HEK293T | ATCC | #CRL-11268, RRID:CVCL_1926 | |
| antibody | Anti-MCAT (mouse monoclonal) | Santa Cruz | #sc-390858, RRID:AB_2827536 | WB: (1:100) |
| Antibody | Anti-OXSM (rabbit polyclonal) | Thermo Fisher Scientific | #PA5-32132, RRID:AB_2549605 | WB: (1:1000) |
| Antibody | Anti-MECR (rabbit polyclonal) | Proteintech | #51027–2-AP, RRID:AB_615013 | WB: (1:1000) |
| Antibody | Anti-Lipoic Acid (rabbit polyclonal) | Abcam | #ab58724, RRID:AB_880635 | WB: (1:1000) |
| Antibody | Anti-DLAT (rabbit monoclonal) | Abcam | #ab172617, RRID:AB_2827534 | WB: (1:1000) |
| antibody | Anti-NDUFAB1 (rabbit polyclonal) | Abcam | #ab96230, RRID:AB_10686984 | WB: (1:1000) |
| Antibody | Anti-Flag (rabbit polyclonal) | Sigma-Aldrich | #F7425, RRID:AB_439687 | WB: (1:1000) |
| Antibody | Anti-V5 (rabbit polyclonal) | Abcam | #ab9116, RRID:AB_307024 | WB: (1:2,000) |
| Antibody | Anti-GRIM19 (mouse monoclonal) | Abcam | #ab110240, RRID:AB_10863178 | WB: (1:1,000) |
| Antibody | Anti-SDHA (mouse monoclonal) | Abcam | #ab14715, RRID:AB_301433 | WB: (1:10,000) |
| Antibody | Anti-UQCRFS1 (mouse monoclonal) | Abcam | #ab14746, RRID:AB_301445 | WB: (1:1000) |
| Antibody | Anti-MTCO1 (mouse monoclonal) | Abcam | #ab14705, RRID:AB_2084810 | WB: (1:1000) |
| Antibody | Anti-ATP5a (mouse monoclonal) | Abcam | #ab14748, RRID:AB_301447 | WB: (1:1000) |
| Antibody | Anti-NDUFA9 (mouse monoclonal) | Abcam | #ab14713, RRID:AB_301431 | WB: (1:1000) |
| Antibody | Anti-SDHB (mouse monoclonal) | Abcam | #ab14714, RRID:AB_301432 | WB:(1:2,000) |
| Antibody | Anti-LYRM4 (rabbit polyclonal) | Thermo-Fisher | #PA5-56448 RRID:AB_2643635 | WB:(0.4 µg/mL) |
| Antibody | Anti-VDAC (rabbit polyclonal) | Cell Signaling | #4866, RRID:AB_2272627 | WB: (1:1000) |
| Antibody | Anti-CS (rabbit polyclonal) | Abcam | #ab96600, RRID:AB_10678258 | WB: (1:1000) |
| Antibody | Anti-Lipoic Acid (rabbit polyclonal) | Millipore | #437695, RRID:AB_212120 | WB: (1:1000) |
| Antibody | Anti-LIPT1 (rabbit polyclonal) | Sigma-Aldrich | #AV48784, RRID:AB_185290 | WB: (1:1000) |
| Antibody | Anti-DLAT (mouse monoclonal) | Cell Signaling | #12362, RRID:AB_279789 | WB: (1:1000) |

*Continued on next page*

*Continued*

| Reagent type (species) or resource | Designation | Source or reference | Identifiers | Additional information |
|---|---|---|---|---|
| Antibody | Anti-DLST (rabbit polyclonal) | Cell Signaling | #5556, RRID:AB_106951 | WB: (1:1000) |
| Antibody | Anti-GAPDH (rabbit monoclonal) | Cell Signaling | #8884 RRID:AB_11129865 | WB: (1:2000) |
| Antibody | Anti-MHC (mouse monoclonal) | DSHB | #SC-71, RRID:AB_2147165 | WB: (0.2 µg/ml) |
| Antibody | Anti-Myogenin (mouse monoclonal) | DSHB | #F5D, RRID:AB_2146602 | WB: (0.5 µg/ml) |
| Antibody | Anti-MyoD (mouse monoclonal) | DSHB | #D7F2, RRID:AB_1157912 | WB: (0.5 µg/ml) |
| Antibody | Goat Anti-Mouse IgG (H and L) Antibody Dylight 800 Conjugated | Rockland | #610-145-002-0.5, RRID:AB_10703265 | WB:(1:10,000) |
| Antibody | Donkey anti-Rabbit IgG (H+L) Highly Cross-Adsorbed Secondary Antibody, Alexa Fluor 680 | Invitrogen | #A10043, RRID:AB_2534018 | WB:(1:10,000) |
| Antibody | Goat Anti-Mouse IgG (H+L) Antibody, Alexa Fluor 680 Conjugated | Invitrogen | #A21057, RRID:AB_141436 | WB:(1:10,000) |
| Antibody | Donkey Anti-Rabbit IgG (H and L) Antibody Dylight 800 Conjugated | Rockland | #611-145-002-0.5, AB_11183542 | WB:(1:10,000) |
| Antibody | Goat anti-Mouse IgG (H+L), Superclonal Recombinant Secondary Antibody, HRP | Thermo Fisher Scientific | #A28177, RRID:AB_2536163 | WB:(1:10,000) |
| recombinant DNA reagent | pMcat CRISPR sgRNA guide 1 (plasmid) | This paper | sgRNA cloned in pLentiCRISPRv2 | GCGCGTCGCAATGAGCGCTC |
| recombinant DNA reagent | pMcat CRISPR sgRNA guide 2 (plasmid) | This paper | sgRNA cloned in pLentiCRISPRv2 | CGAGGCCGCGCACCGGGTAC |
| recombinant DNA reagent | pOxsm CRISPR sgRNA guide 1 (plasmid) | This paper | sgRNA cloned in pLentiCRISPRv2 | CTAGTGACACCACTTGGCGT |
| recombinant DNA reagent | pOxsm CRISPR sgRNA guide 2 (plasmid) | This paper | sgRNA cloned in pLentiCRISPRv2 | CGTTGGGACTCAACTAGTTT |
| recombinant DNA reagent | pMecr CRISPR sgRNA guide 1 (plasmid) | This paper | sgRNA cloned in pLentiCRISPRv2 | AGGCTTGGTACCGCCACGGC |
| recombinant DNA reagent | pMecr CRISPR sgRNA guide 2 (plasmid) | This paper | sgRNA cloned in pLentiCRISPRv2 | CGTGGCGGTAC CAAGCCTCG |
| recombinant DNA reagent | pLipt1 CRISPR sgRNA guide 1 (plasmid) | This paper | sgRNA cloned in pLentiCRISPRv2 | CACCGCTTCTAGA TGTATGTGGTCG |
| recombinant DNA reagent | Lipt1 CRISPR sgRNA guide 2 (plasmid) | This paper | sgRNA cloned in pLentiCRISPRv2 | CACCGCTCCTTCTG TCGTCATCGGC |

*Continued on next page*

*Continued*

| Reagent type (species) or resource | Designation | Source or reference | Identifiers | Additional information |
|---|---|---|---|---|
| recombinant DNA reagent | pLentiCRISPRv2 (plasmid) | Addgene | #52961 RRID:Addgene_52961 | |
| recombinant DNA reagent | pLKO.1 shFASN (plasmid) | The Broad Institute | #shRNA TRCN0000075703 | |
| recombinant DNA reagent | pshScramble (plasmid) | Addgene | #1864, RRID:Addgene_1864 | |
| recombinant DNA reagent | psPAX2 (plasmid) | Addgene | #12259, RRID:Addgene_12259 | |
| recombinant DNA reagent | pDM2.G (plasmid) | Addgene | #12260, RRID:Addgene_12260 | |
| recombinant DNA reagent | pQXCIP mtDSRed (plasmid) | This paper | | mtDSRed in pQXCIP backbone |
| recombinant DNA reagent | pQXCIP-Δ4, 5CMV-mMcat (plasmid) | This paper | | Mouse Mcat with truncated CMV promoter in pQXCIP backbone |
| recombinant DNA reagent | pQXCIP-CMV-mMcat (plasmid) | This paper | | Mouse Mcat with full CMV promoter in pQXCIP backbone |
| recombinant DNA reagent | pQXCIP-CMV-mOxsm (plasmid) | This paper | | Mouse Oxsm with full CMV promoter in pQXCIP backbone |
| recombinant DNA reagent | pQXCIP-Δ4, 5CMV-mMecr (plasmid) | This paper | | Mouse Mecr with truncated CMV promoter in pQXCIP backbone |
| recombinant DNA reagent | pQXCIP-CMV-mMecr (plasmid) | This paper | | Mouse Mecr with full CMV promoter in pQXCIP backbone |
| sequence-based reagent | Mcat CRISPR Ver Fwd | This paper | PCR primers | GACCGACATGCAACTGCAAATAG |
| sequence-based reagent | Mcat CRISPR Ver Rev | This paper | PCR primers | GGCCAGTGAAGCCACAAAGA |
| sequence-based reagent | Oxsm CRISPR Ver Fwd | This paper | PCR primers | CAACCATGTTGTCAAAATGCTTG |
| sequence-based reagent | Oxsm CRISPR Ver Rev | This paper | PCR primers | GGTCTGAAACAGCAAAGCAGTTTC |
| sequence-based reagent | Mecr CRISPR Ver Fwd | This paper | PCR primers | GCTGTCGCGGACGAATG |
| sequence-based reagent | Mecr CRISPR Ver Rev | This paper | PCR primers | GTCGGAAGCATCCACTGAGAC |
| commercial assay or kit | TruSeq Stranded mRNA Library Prep kit | Illumina | #20020595 | |
| commercial assay or kit | TruSeq RNA UD Indexes | Illumina | #20022371 | |
| commercial assay or kit | NovaSeq S1 reagent Kit | Illumina | #20027465 | |
| commercial assay or kit | Kapa Library Quant Kit | Kapa Biosystems | #KK4824 | |
| commercial assay or kit | Pierce BCA Assay | Thermo | #23225 | |

*Continued on next page*

*Continued*

| Reagent type (species) or resource | Designation | Source or reference | Identifiers | Additional information |
|---|---|---|---|---|
| commercial assay or kit | D1000 ScreenTape assay | Agilent Technologies | 5067–5583 | |
| chemical compound, drug | [U-$^{13}$C]glutamine | Cambridge Isotopes | #CLM-1822 | |
| chemical compound, drug | [U-$^{13}$C]glucose | Cambridge Isotopes | #CLM-1396 | |
| chemical compound, drug | Digitonin | GoldBio | #D-180–2.5 | |
| chemical compound, drug | SPLASH Lipidomix | Avanti Polar Lipids | #330707 | |
| software, algorithm | Agilent Mass Hunter Qual B.07.00 | Agilent | https://www.agilent.com/en/products/software-informatics/masshunter-suite/masshunter/masshunter-software | |
| software, algorithm | Agilent Mass Hunter Quant B.07.00 | Agilent | https://www.agilent.com/en/products/software-informatics/masshunter-suite/masshunter/masshunter-software | |
| software, algorithm | Lipid Annotator | Agilent | https://www.agilent.com/en/products/software-informatics/mass-spectrometry-software/data-analysis/mass-profiler-professional-software | |
| software, algorithm | MetaboAnalyst 4.0 | *Xia and Wishart, 2016* | http://www.metaboanalyst.ca | |
| Other | Lipofectamine 2000 transfection reagent | Invitrogen | 11668019 | |

## *Mcat*, *Oxsm*, and *Mecr* mutant cell lines

Unless otherwise indicated for specific experiments, C2C12 immortalized mouse skeletal myoblasts (ATCC CRL-1772, verification provided by ATCC, mycoplasma testing status: negative) were grown in DMEM with 4.5 g/L glucose, glutamine, and sodium pyruvate (Corning, 10–013-CV) supplemented with 10% FBS (Sigma, F0926) at 37°C in a humidified atmosphere containing 5% $CO_2$. To generate mtFAS knockout clones, two guides per gene were designed targeting exon 1 or exon 2 of the mouse *Mcat*, *Oxsm*, and *Mecr* genes (see key resources table for sequences) and cloned in the pLentiCRISPR v2 vector. Parental C2C12 cells were transfected with pLentiCRISPRv2_sgmtFAS or empty vector. Editing efficiency was assessed in the bulk transfected population by T7E1 assay. Briefly, DNA was isolated using QuickExtract solution (Lucigen). Edited regions were PCR amplified with the primers listed in the key resources table. PCR products were denatured, reannealed, and digested with T7E1 endonuclease (New England Biologicals), which cleaves mismatched DNA. Digests were run on a 2% agarose gel, visualized with SYBR safe DNA stain (Invitrogen) and percent editing was calculated as the sum of the cleaved fragments divided by the total DNA in each lane. To establish clonal lines, 48 hr after transfection, the cells were sorted based on GFP signal and plated in a 96-well plate to obtain single cell colonies. Colony number and size was assessed by brightfield microscopy. Clones were screened based on MCAT, OXSM, and MECR protein levels. Three *Oxsm* and *Mecr* mutant clones, three control clones, and two *Mcat* mutant clones were ultimately selected for

further experiments. Quantification of single cell clone number and colony size was assessed 7 days after single cell sort. Small clones = <150 cells, Med = 150–500 cells, Large = >500 cells. For growth assays, cells were plated at 100,000 cells/plate in 10 cm dishes in 4.5 g/L glucose and counted every 24 hr using a BioRad TC20 automated cell counter. For rescue experiments, control, Mcat, Oxsm, and Mecr mutant clonal cell lines were infected with retrovirus harboring a control transfer plasmid (pQXCIP mtDSRed, pCtrl) or pQXCIP expressing the indicated mtFAS gene off the full CMV (pQXCIP-CMV-Oxsm, pOxsm, pCMV-Mcat, pCMV-Mecr) or Δ4,5 truncated CMV promoter (pΔ4,5CMV-Mcat, pΔ4,5CMV- Mecr). mtFAS genes were cloned from mouse cDNA isolated from NIH3T3L1 cells. Stably infected cells were selected with 2 µg/mL puromycin for 7 days. 10,000 cells/ well were plated in 6-well or 12-well plates, and grown in normal growth medium containing 4.5 g/L glucose or 10 mM glutamate as indicated for 3 days (glucose) or 4 days (galactose) until control wells reached confluency. Plates were stained for 10 min in 2.5 mg/mL crystal violet in 20% methanol, washed 6x and allowed to dry, then imaged.

## Stable FASN knockdown

HEK293T cells were transiently transfected with pLKO.1 shFASN or scramble shRNA control along with the lentiviral packaging plasmids psPAX2 and pMD2.G using polyethylenimine. 48 hr after transfection, viral supernatant was collected, filtered through a 0.45 µm polyethersulfone membrane, and stored at 4°C. 10 µg/mL Polybrene (EMD Millipore, TR-1003-G) was added and a 1:1 mixture of viral supernatant and fresh growth medium (DMEM + 10%FBS) was applied directly to *Oxsm* mutant cells, along with wild type C2C12 controls, and incubated for 16 hr at 37°C in a humidified incubator with 5% $CO_2$. Viral medium was discarded and replaced with fresh growth medium and cells were allowed to recover and expand for 24 hr. After the recovery period, stably infected cells were selected with 2 µg/mL puromycin for 1 week. Knockdown of FASN was confirmed via immunoblotting as described below.

## *Lipt1* mutant cell lines

C2C12 myoblasts were grown in DMEM High Glucose (Sigma, D5796) supplemented with 10% FBS (Gemini Bio-Products #100–106) and penicillin/streptomycin (HyClone) at 37°C in a humidified atmosphere containing 5% $CO_2$. To establish LIPT1 mutant clones, two guides were designed targeting the coding region of the mouse *Lipt1* gene and cloned in the px459 vector. Parental C2C12 cells were transfected with px459_sgLIPT1 or empty vector and three days later the cells were sorted based on GFP signal. GFP positive cells were subsequently plated to obtain single cell colonies and screened based on LIPT1 protein levels and lipoylation of pyruvate dehydrogenase (PDH) and 2-oxo-glutarate dehydrogenase (OGDH). Two *Lipt1* mutant clones and one wild type clone were ultimately selected for further experiments.

## Crude mitochondrial isolation

Cells were harvested by incubation with 0.25% trypsin-EDTA (Gibco, 25200–072), pelleted, washed once with ice cold sterile PBS (Gibco, 10010–023), and frozen at −80°C. Upon thawing, cells were resuspended in 1 mL CP-1 buffer (100 mM KCl, 50 mM Tris-HCl, 2 mM EGTA, pH 7.4) and mechanically lysed by 10 passes through a 27-gauge needle, and centrifuged at 700 x g to pellet unlysed cells and debris. Supernatant was moved to a new tube and centrifuged at 10,000 x g to pellet crude mitochondrial fraction. Post-mitochondrial supernatant (PMS) was saved for immunoblotting or discarded. Mitochondrial pellets were resuspended in a small volume of CP-1 buffer equal to approximate pellet volume and used in assays described below.

## SDS-PAGE and immunoblotting

Whole cell lysates were prepared by scraping cells directly into Ripa buffer (50 mM Tris-HCl, 1% NP-40, 0.5% Na-Deoxycholate, 0.1% SDS, 150 mM NaCl, 2 mM EDTA) supplemented with protease and phosphatase inhibitors (Sigma Aldrich P8340, Roche Molecular 04906845001), incubated on ice for 45 min with vortexing every 5 min, and then centrifuged at 16,000 x g for 10 min at 4°C to remove insoluble material. Supernatant was saved as whole cell lysate (WCL). WCL, PMS, and/or crude mitochondrial fractions were normalized for total protein content via BCA Assay (Thermo Scientific 23225). Samples were resolved by SDS-PAGE on Tris-glycine gels (Invitrogen XP04205BOX) and

transferred to nitrocellulose or PVDF membranes. Immunoblotting was performed using the indicated primary antibodies which are listed in the key resources table according to the manufacturers' recommendations, and analyzed by Licor Odyssey.

## Blue Native-PAGE

Crude mitochondrial fractions were isolated and normalized for total protein content as described above. Blue Native (BN) PAGE was performed using the Invitrogen NativePAGE system. 100 µg of mitochondria were pelleted at 10,000 x g for 10 min at 4°C and resuspended in 1x pink lysis buffer (Invitrogen, BN20032). Digitonin (GoldBio D-180–2.5) was added to a final concentration of 1% mass/volume. Samples were incubated on ice for 15 min, then spun for 20 min at 20,000 x g. 6 µL of NativePAGE sample buffer (Invitrogen, BN20041) was added and 10 µL of sample was run on pre-cast 3–12% NativePAGE gels (Invitrogen, BN2011B × 10) with NativePAGE anode buffer (Invitrogen, BN2001) and dark blue cathode buffer (Invitrogen, BN2002) at 150 V for 1 hr then switched to light blue cathode buffer (Invitrogen, BN2002) and run at 30 V overnight. Gels were subsequently transferred to PVDF at 100 V, washed with methanol, and blotted with the indicated primary antibodies which are listed in the key resources table according to the manufacturers' recommendations. Secondary anti-mouse HRP antibody listed in the key resources table and SuperSignal West Femto Maximum Sensitivity Substrate (Thermo, 34096) was used to visualize bands on film (GeneMate, F-9024−8 × 10).

## Flux lipidomics

Quadruplicate biological replicates of mtFAS mutant and control cells were plated at 35% confluence in normal growth medium, then switched to labeling medium (DMEM base (Corning, 17–207-CV), supplemented with 25 mM U$^{13}$C-glucose (Cambridge Isotopes, CLM-1396) and 4 mM glutamine (Sigma G7513)) for 0, 0.25, 0.5, or one population doubling time. Samples were harvested at 70% confluence via incubation with 0.25% trypsin (Gibco, 25200–072), and pelleted at 300 x g for 5 min at 4°C, flash frozen in liquid nitrogen, and stored at −80°C. Lipids were extracted from cell pellets in a randomized order using a solution of 225 µL methanol (MeOH) containing internal standards (IS) (Avanti SPLASH LIPIDOMIX, 10 µL each/sample) and 750 µL methyl tert-butyl ether (MTBE) (*Matyash et al., 2008*). The samples were sonicated for 1 min, rested on ice for 1 hr, briefly vortexed every 15 min then 188 µL dd-H$_2$O was added to induce phase separation. All solutions were pre-chilled on ice. The sample was then vortexed for 20 s, rested at room temperature for 10 min, and centrifuged at 3000 x g for 5 min at 4°C. The upper, organic phases were collected and evaporated to dryness under vacuum. Lipid samples were reconstituted in 250 µL isopropyl alcohol. Concurrently a process blank sample was brought forward as well as a pooled quality control (QC) sample, prepared by taking equal volumes (~10 µL per sample) from each sample after final resuspension. Lipid extracts were separated on a Waters Acquity UPLC CSH C18 1.7 µm 2.1 × 100 mm column maintained at 65°C connected to an Agilent HiP 1290 Sampler, Agilent 1290 Infinity pump and Agilent 6530 Accurate Mass Q-TOF dual AJS-ESI mass spectrometer. For positive mode, the source gas temperature was set to 225°C, with a drying gas flow of 11 L/min, nebulizer pressure of 40 psig, sheath gas temp of 350°C and sheath gas flow of 11 L/min. VCap voltage was set at 3500 V, nozzle voltage 1000 V, fragmentor at 110 V, skimmer at 85 V and octopole RF peak at 750 V. For negative mode, the source gas temperature was set to 300°C, with a drying gas flow of 11 L/min, a nebulizer pressure of 30 psig, sheath gas temp of 350°C and sheath gas flow 11 L/min. VCap voltage was set at 3500 V, nozzle voltage 2000 V, fragmentor at 100 V, skimmer at 65 V and octopole RF peak at 750 V. Samples were analyzed in a randomized order in both positive and negative ionization modes in separate experiments acquiring with the scan range *m/z* 100–1700. Mobile phase A consists of ACN:H$_2$O (60:40 *v/v*) in 10 mM ammonium formate and 0.1% formic acid, and mobile phase B consists of IPA:ACN:H$_2$O (90:9:1 *v/v*) in 10 mM ammonium formate and 0.1% formic acid. The chromatography gradient for both positive and negative modes starts at 15% mobile phase B then increases to 30% B over 2.4 min, it then increases to 48% B from 2.4 to 3.0 min, then increases to 82% B from 3 to 13.2 min, then increases to 99% B from 13.2 to 13.8 min where it was held until 16.7 min and then returned to the initial conditions and equilibrated for 5 min. Flow was 0.4 mL/min throughout, injection volume was 3 µL for positive and 10 µL negative mode. Tandem mass spectrometry was conducted using the same LC gradient at collision energy of 25 V. QC samples (n = 8)

and blanks (n = 4) were injected throughout the sample queue and ensure the reliability of acquired data. Results from LC-MS experiments were collected using Agilent Mass Hunter (MH) Workstation and analyzed using the software packages MH Qual, MH Quant, and Lipid Annotator (Agilent Technologies, Inc).

## FAMES fluxomics

Lipid extracts were saponified by an addition of 1 mL of 5% HCl in MeOH. Samples were vortexed for 15 seconds then placed into a sand bath heated at 80℃ for 2 hours. The resulting methanolysis reaction products were extracted with hexanes (3 X 1 mL) by vortexing for 15 seconds and then centrifuged for 4 minutes at 10,000 x g. Supernatants were transferred to new tubes and gently dried under a nitrogen stream at room temperature. Samples were resuspended in 200 µL EtOAc and transferred to GC-MS vials with insert. GC-MS analysis was performed with an Agilent 5977b GC-MS HES-MSD and an Agilent 7693A automatic liquid sampler. Samples were injected (1 µL) 10:1, with an inlet temperature of 250℃. The gas chromatograph had an initial temperature of 67℃ for one minute followed by a 50℃/min ramp to 197℃, a second ramp of 10℃/minute up to 325℃ began immediately with a hold time of 2 minutes. A 30-meter Agilent Zorbax DB-5MS with 10 m Duraguard capillary column was employed for chromatographic separation. Helium was used as the carrier gas at a rate of 1 mL/minute. A gain of 5.0 was used for data acquisition. High Efficiency Source and quadrupole temperatures were held at 275 ℃ and 190 ℃, respectively.

## Kinetics (Fluxomics) calculation

Target compounds were fed to the emass software created by the Rockwood group (*Rockwood and Haimi, 2006*). This software calculated masses and relative abundances for all isotopes observed so if for one compound only M2 was observed and another went out to M5, emass would calculate M0-M2 for compound 1 and M0-M5 for compound 2. Matrixes of the theoretical isotopic abundances and the measured isotopic abundances were created. The multiplicative inverse of each matrix was calculated and then the dot product of the theoretical inverse matrix and the measured inverse matrix was calculated. The data from one row of the resulting matrix (all rows are identical) is then sum normalized. In an unlabeled sample M0 will be measured as 1 and all other isotopes should be 0 (standard Metabolomic Flux Analysis).

## Cytochrome $bc_1$ complex (CIII) and citrate synthase (CS) activity assays

CIII and CS activities were measured following the protocol published in *Spinazzi et al., 2012*. For CIII activity assay, 10 µg of purified mitochondria was mixed in 990 µL of the assay buffer (25 mM potassium phosphate (pH 7.5), 75 µM oxidized cytochrome c, 500 µM KCN, 100 µM EDTA (pH 7.5), 0.025% (volume/volume) Tween-20). After the baseline reading at 550 nM for 2 min, the reaction was started by adding 10 µl of 10 mM decylubiquinol and the increasing absorbance at 550 nm was measured for the next 3 min. In parallel CIII-independent cytochrome c reduction was measured by adding antimycin A in a separate reaction mixture and subtracted from the reaction without antimycin A. CIII activity was calculated and expressed as nmole/min/mg. CS activity was measured by incubating 5 µg of purified mitochondria in 950 µL of the assay buffer (100 mM Tris (pH 8.0), 0.1% (volume/volume) Triton X-100, 100 µM DTNB (5,5'-dithiobis(2-nitrobenzoic acid)), 300 µM Acetyl CoA). After reading the baseline activity at 412 nm for 2 min, the reaction was started by adding 50 µL of 10 mM oxaloacetic acid (OA) and monitored the increase in the absorbance at 412 nm for 3 min. CS activity was calculated by subtracting the baseline increase at 412 nm from the increase after addition of OA and expressed as nmole/min/mg.

## Seahorse experiments

Seahorse experiments were performed on a Seahorse XFe96 Analyzer. Cells from n = 2 clones (Mcat mutants) or n = 3 clones (Oxsm, Mecr mutants and controls) were plated in 8 wells of a 96-well seahorse plate at a density of 6,000 cells/well in DMEM + 10% FBS. The morning of the experiment, cells were washed 2x and then medium was replaced with DMEM (Sigma D5030) supplemented with 25 mM glucose, 2 mM glutamine and 1 mM pyruvate, pH 7.4. A standard mitochondrial stress test was performed using 1 µM oligomycin, 3 µM FCCP, and 0.5 µM Rotenone + 0.5 µM Antimycin A. Assay protocol was standard (3 measurements per phase, acute injection followed by 3 min mixing,

0 min waiting, and 3 min measuring). Data were normalized to total cellular protein per well (Thermo BCA Kit cat. 23225). Results were analyzed in WAVE software and processed through the XF Mito Stress Test Report.

## Mitochondrial membrane potential microscopy and quantitation

Cells were plated in chambered coverglass (Fisher, 155409) and allowed to adhere overnight. Medium was changed and replaced with fresh DMEM + 10% FBS containing 25 nM MitoTracker Red CMXROS and 50 nM MitoTracker Green (Invitrogen M7512 and M7514). Cells were incubated with dyes for 30 min at 37°C in a humidified incubator with 5% CO2, then washed twice and replaced with fresh medium, and imaged using a Zeiss AxioObserver Z1 fluorescence microscope (Carl Zeiss) equipped with 40x oil-immersion objective. Digital fluorescence and differential interference contrast (DIC) images were acquired using a monochrome digital camera (AxioCam 506, Carl Zeiss). Images were exported from Zeiss Zen software package and analyzed for fluorescence intensity using FiJi ImageJ. Images are representative of n $\geq$ 38 images from two clones imaged in two independent experiments.

## Co-immunoprecipitation of ACP-Flag and LYRM-V5 proteins

10 cm plates of HEK293T cells were transiently transfected with 10 µg of total DNA using Lipofect-amine 2000 (Invitrogen, 11668019). DNA was made up of equal amounts of empty vector, pcDNA3.1 hNDUFAB1-Flag, and pcDNA3.1 LYRM-V5 constructs alone or in combination as indicated. 48 hr after transfection, cells were collected by incubation with 0.25% trypsin, washed, and pelleted at 300 g. Crude mitochondria were isolated as described above and protein content was assessed via BCA assay. 500 µg of crude isolated mitochondria were pelleted and resuspended in 500 µL of XWA buffer (20 mM HEPES, 10 mM KCl, 1.5 mM MgCl2, 1 mM EDTA, 1 mM EGTA, 150 mM NaCl, 2.1 mg/mL NaF, 10.8 mg/mL β-glycerophosphate, pH 7.4) then lysed by incubation in 0.7% digitonin for 30 min rotating at 4°C. Insoluble material was removed by centrifugation at 20,000 g for 20 min, and supernatant was incubated with 20 µL anti-Flag agarose beads (Sigma, F2426) for 2 hr rotating at 4°C. Beads were washed 3 times with XWA buffer and samples were eluted in 40 µL of 1x Lamelli buffer (10% glycerol, 50 mM Tris HCl, pH 6.8, 1% SDS) for 10 min at 65°C. Samples were loaded on Tris-Glycine gels, separated by SDS-PAGE, and immunoblotted for Flag and V5 as described above. Results are representative of three separate experiments.

## Quantitative proteomics

Duplicate biological samples of two control clones, two *Oxsm* mutant clones, and two *Mecr* mutant clones were grown in standard proliferative conditions to 70% confluence and harvested by incubation with 0.25% trypsin, pelleted at 300 g, washed with sterile PBS, flash frozen in liquid nitrogen and stored at −80°C. As only eleven samples could be run with the SL-TMT protocol, one *Oxsm* clone had only one sample analyzed and was ultimately omitted from the results.

### Protein extraction, digestion, and TMT labeling

Samples were processed using the SL-TMT protocol as described previously (*Navarrete-Perea et al., 2018*). Specifically, mitochondrial pellets were resuspended in 200 µL of lysis buffer (8M urea, 200 mM EPPS pH 8.5 plus 1X cOmplete protease and phosphatase inhibitor) and syringe lysed (10 times with 21-gauge needle). Following a BCA assay to estimate protein concentration, all lysates were reduced (20 min with 5 mM TCEP at room temperature), alkylated (20 min with 10 mM iodoacetamide, at room temperature in the dark), and quenched (20 min with 10 mM DTT, at room temperature in the dark). Proteins were precipitated by chloroform methanol precipitation, as described previously (*Paulo and Gygi, 2015*). Precipitated proteins were resuspended in 200 mM EPPS pH 8.5 (~1 mg/ml) and digested first with LysC (Wako) for 3 hr at 37°C shaking on a vortexer (speed = 50%) followed by a 6 hr trypsin digestion at 37°C (sequencing grade, Pierce). Both enzymes were added in the presence of beads and at a 1:100 protease-to-peptide ratio. Following digestion, the samples were centrifuged as above and held to the magnet for 2 min. Digested peptides were simply transferred into a new tube. The beads were then washed with 50 µL of 0.2M EPPS pH8.5, which was combined with the initial elution. We added a final volume of 30% acetonitrile to the eluted peptides and labelled the 50 µg of peptide with 100 µg of TMT directly into the digestion

mixture. To check mixing ratios, 2 µg of each sample were pooled, desalted, and analyzed by mass spectrometry. Using normalization factors calculated from this 'label check,' samples were mixed 1:1 across all 11 channels and desalted using a 50 mg Sep-Pak solid phase extraction column. The approximately 300 µg of peptide were fractionated with basic pH reversed-phase (BPRP) HPLC, collected in a 96-well plate and combined down to 24 fractions prior to desalting and subsequent LC-MS/MS processing (*Paulo, 2014*; *Paulo and Gygi, 2017*).

## Mass spectrometry analysis

Mass spectrometric data were collected on an Orbitrap Fusion mass spectrometer coupled to a Proxeon NanoLC-1000 UHPLC. The 100 µm capillary column was packed with 35 cm of Accucore 150 resin (2.6 µm, 150 Å; ThermoFisher Scientific). The SPS-MS3 method use used to reduce ion interferences that may result in ratio compression (*Gygi et al., 2019*; *Paulo et al., 2016*). Peptides in each fraction were separated using a 150 min gradient from ~ 5% to 35% acetonitrile. The scan sequence began with an MS1 spectrum (Orbitrap analysis, resolution 120,000, 350–1400 Th, automatic gain control (AGC) target 5E5, maximum injection time 100 ms). The top ten precursors were then selected for MS2/MS3 analysis. MS2 analysis consisted of collision-induced dissociation (CID), quadrupole ion trap analysis, automatic gain control (AGC) 2E4, NCE (normalized collision energy) 35, q-value 0.25, maximum injection time 120 ms), and isolation window at 0.7. Following acquisition of each MS2 spectrum, we collected an MS3 spectrum in which multiple MS2 fragment ions are captured in the MS3 precursor population using isolation waveforms with multiple frequency notches. MS3 precursors were fragmented by HCD and analyzed using the Orbitrap (NCE 65, AGC 1.5E5, maximum injection time 150 ms, resolution was 50,000 at 400 Th). Spectra were converted to mzXML via MSconvert (*Chambers et al., 2012*). Database searching included all entries from the *Mus musculus* Database (UniProt; August 2017). The database was concatenated with one composed of all protein sequences for that database in the reversed order. Searches were performed using a 50-ppm precursor ion tolerance for total protein level profiling. The product ion tolerance was set to 0.9 Da. These wide mass tolerance windows were chosen to maximize sensitivity in conjunction with SEQUEST searches and linear discriminant analysis (*Beausoleil et al., 2006*; *Huttlin et al., 2010*). TMT tags on lysine residues and peptide N termini (+229.163 Da) and carbamidomethylation of cysteine residues (+57.021 Da) were set as static modifications, while oxidation of methionine residues (+15.995 Da) was set as a variable modification. Peptide-spectrum matches (PSMs) were adjusted to a 1% false discovery rate (FDR) (*Elias and Gygi, 2007*; *Elias and Gygi, 2010*). PSM filtering was performed using a linear discriminant analysis, as described previously (*Huttlin et al., 2010*) and then assembled further to a final protein-level FDR of 1% (*Elias and Gygi, 2007*). Proteins were quantified by summing reporter ion counts across all matching PSMs, also as described previously (*McAlister et al., 2012*). Reporter ion intensities were adjusted to correct for the isotopic impurities of the different TMT reagents according to manufacturer specifications. The signal-to-noise (S/N) measurements of peptides assigned to each protein were summed and these values were normalized so that the sum of the signal for all proteins in each channel was equivalent to account for equal protein loading. Finally, each protein abundance measurement was scaled, such that the summed signal-to-noise for that protein across all channels equals 100, thereby generating a relative abundance (RA) measurement.

Technical replicates from the geometric mean-normalized proteomics data were averaged. Heatmaps were generated by calculating the average value for the GFP replicates and dividing each replicate (experimental and control) by this value for each protein. A list of ETC component proteins was passed into the xpressplot.heatmap function from XPRESSplot (*Berg et al., 2020*; *Waskom et al., 2018*) and further formatting was performed using Matplotlib (*Hunter, 2007*). The volcano plot for the Mecr vs GFP proteomics data was generated by providing lists of Mitocarta (*Calvo et al., 2016*; *Pagliarini et al., 2008*) and ETC proteins and the technical-replicate, geometric-normalized dataframe for both GFP and Mecr replicates to the xpressplot.volcano function (*Berg et al., 2020*).

## *Lipt1* KO seahorse experiments

Seahorse experiments were performed on a Seahorse XFe96 Analyzer. Cells were plated at a density of 10,000 cells/well in DMEM + 10% FBS. The morning of the experiment, cells were washed 2x and

then medium was replaced with Seahorse medium: DMEM (Sigma D5030) supplemented with 10 mM glucose, 2 mM glutamine and 1 mM pyruvate, pH 7.4. A standard mitochondrial stress test was performed using 2 µM oligomycin, 1 µM Carbonyl cyanide m-chlorophenyl hydrazone (CCCP), and then 1 µM Antimycin A. Assay protocol was standard (3 measurements per phase, acute injection followed by 3 min mixing, 0 min waiting, and 3 min measuring). Data were normalized to 10,000 cells/well. Results were analyzed in WAVE software and processed through the XF Mito Stress Test Report.

## Glutamine tracing experiments

C2C12 cells were plated in a 60 mm plate at a density of 100,000 cells/plate. The following day, medium was changed to tracing medium: DMEM (Sigma 5030) supplemented with 2.5 mM glucose, 4 mM [U-$^{13}$C]glutamine, 1 mM pyruvate, and 5% dialyzed FBS. Cells were harvested 24 hr later by washing once with ice-cold normal saline and then quenched with 800 µL 80:20 methanol:water. Methanol lysates were then subjected to three rapid freeze-thaw cycle and then spun at 16,000 x g for 10 min at 4°C. The supernatants, with 1 µL d27-myristic acid added (Sigma 68698) (GCMS only) as an internal control, were evaporated using a SpeedVac concentrator.

## Liquid chromatography mass spectrometry (LCMS)

A replicate plate was set up during tracing experiments to assess relative metabolite concentrations using LCMS and harvested as described above. Dried metabolites were reconstituted in 100 µl of 0.03% formic acid in analytical-grade water, vortexed and centrifuged to remove insoluble material. The supernatant was collected and subjected to targeted metabolomics analysis as described on an AB SCIEX QTRAP 5500 liquid chromatography/triple quadrupole mass spectrometer (Applied Biosystems SCIEX) (*Kim et al., 2000*). The injection volume was 20 µl. Chromatogram review and peak area integration were performed using MultiQuant (version 2.1, Applied Biosystems SCIEX). The peak area for each detected metabolite was normalized against the total ion count of that sample.

## Gas chromatography mass spectrometry (GCMS) derivatization

The supernatants, with 1 µL d27-myristic acid added as an internal control, were evaporated using a speed vac. Dried metabolites were re-suspended in 30 µL anhydrous pyridine with 10 mg/mL methoxyamine hydrochloride and incubated at room temperature overnight. The following morning, the samples were cooked at 70°C for 10–15 min and then centrifuged at 16,000 x g for 10 min. The supernatant was transferred to a pre-prepared GC/MS autoinjector vial containing 70 µL N-(tert-butyldimethylsilyl)-N-methyltrifluoroacetamide (MTBSTFA) derivatization reagent. The samples were incubated at 70°C for 1 hr following which aliquots of 1 µL were injected for analysis. Samples were analyzed using either an Agilent 6890 or 7890 gas chromatograph coupled to an Agilent 5973N or 5975C Mass Selective Detector, respectively. The observed distributions of mass isotopologues were corrected for natural abundance.

## C2C12 differentiation experiments

MtFAS mutants and controls were seeded in 10 cm plates and grown in DMEM (Corning, 10–013-CV) supplemented with 10% FBS (Sigma, F0926) to 70% confluency (Day −1). The next day at 95% confluency cells were switched to DMEM supplemented with 2% horse serum (Thermo, 16050122) (Day 0). For the next 7 days, medium was changed and replaced with fresh DMEM + 2% horse serum every 24 hr. Samples were imaged via brightfield microscopy or harvested for immunoblotting or RNAseq experiments at the indicated time points after switching to differentiation medium.

## RNA-Seq

### RNA isolation and quality assessment

Four replicates of each genotype (GFP, *Mcat*, *Oxsm,* and *Mecr* mutants) were grown and harvested at Day −1, Day 0, and Day 1 of the differentiation time course as described above —with the exception of GFP Day 0, which only had three replicates. Total RNA was extracted from one confluent 10 cm dish per replicate using TRI Reagent (Zymo Research, R2050-1-200) according to the manufacturer's protocol. Briefly, cells were homogenized in 1 ml of TRI reagent for five mins. After addition

of 0.2 mL of chloroform, the homogenate was shaken vigorously for 15 s, allowed to stand for 15 mins at RT and centrifuged at 12,000xg for 15 min at 4°C. The aqueous phase was mixed with 0.5 mL of 2-propanol and 10 µg of glycogen (Thermo Scientific, R0561) and RNA was precipitated overnight in −20°C. The precipitated RNA was pelleted at 15,000xg for 30 min at 4°C, washed with 1 mL of 75% EtOH and dissolved in nuclease-free water. RNA quantity and purity were assessed using the NanoDrop One spectrophotometer (Thermo Fisher) and RNA integrity was assessed using the Agilent RNA ScreenTape Assay.

### Illumina TruSeq Stranded mRNA Library Prep (RIN 8–10) with UDI

Intact poly(A) RNA was purified from total RNA samples (100–500 ng) with oligo(dT) magnetic beads and stranded mRNA sequencing libraries were prepared as described using the Illumina TruSeq Stranded mRNA Library Prep kit (20020595) and TruSeq RNA UD Indexes (20022371). Purified libraries were qualified on an Agilent Technologies 2200 TapeStation using a D1000 ScreenTape assay (cat# 5067–5582 and 5067–5583). The molarity of adapter-modified molecules was defined by quantitative PCR using the Kapa Biosystems Kapa Library Quant Kit (cat#KK4824). Individual libraries were normalized to 1.30 nM in preparation for Illumina sequence analysis. NovaSeq 2 × 50 bp Sequencing: Sequencing libraries (1.3 nM) were chemically denatured and applied to an Illumina NovaSeq flow cell using the NovaSeq XP chemistry workflow (20021664). Following transfer of the flowcell to an Illumina NovaSeq instrument, a 2 × 51 cycle paired end sequence run was performed using a NovaSeq S1 reagent Kit (20027465). Data Analysis: Reads were trimmed of adaptor sequence using cutadapt (v.2.8) (*Martin, 2011*). Trimmed reads were aligned to the mouse genome (mm10) using STAR (v.2.5) (*Dobin et al., 2013*) with the parameters '-outFilterMismatchNoverLmax 0.2 –sjdbOverhang 49 –outReadsUnmapped Fastx –outSAMtype BAM SortedByCoordinate –quantMode GeneCounts' and with '-sjdbGTFfile' supplied with transcript annotations from Ensembl (Mus_musculus.GRCm38.99.chr_patch_hapl_scaff.gtf). Three samples were censored from further analyses (one replicate each of GFP day minus one, Mcat day zero, and Oxsm day one) due to outlier status relative to all remaining samples. Differential expression was determined using DESeq2 (v.1.26.0) (*Love et al., 2014*). Count data were shrunk using the 'apeglm' option of the DESeq2 function 'lfcShrink'.

## Quantification and statistical analyses

For flux lipidomics and metabolomics, mitoTracker quantification, and Seahorse cellular respiration, statistically significant differences were determined using GraphPad Prism 8. Data were analyzed by one-way or two-way Anova as appropriate followed by Dunnett's multiple comparison test (when compared only to control) or Tukey's multiple comparison test (when comparing all groups, such as for lipid isotopologue analysis). A p-value of 0.05 was considered to be statistically significant. For RNAseq and quantitative proteomics, analyses are described in the corresponding sections.

## Resource availability

All unique/stable reagents generated within this study are available from the corresponding author, Jared Rutter (rutter@biochem.utah.edu), upon request without restriction.

## Data and code availability

The data have been deposited to the ProteomeXchange Consortium via the PRIDE (*Perez-Riverol et al., 2019*) partner repository. The code used to process these data is available at Github https://github.com/j-berg/nowinski_2020 (copy archived at https://github.com/elifesciences-publications/nowinski_2020). Sequencing data have been deposited at the Gene Expression Omnibus (http://www.ncbi.nlm.nih.gov/geo) under accession number GSE148617.

## Acknowledgements

The authors are grateful to members of the Department of Biochemistry at the University of Utah for useful discussion and feedback, specifically Chintan Kikani, Gabrielle Kardon, and Dana Carroll. Cloning of CRISPR plasmids was done by the Mutation Generation and Detection Core at the University of Utah. FACS to generate single cell clones was done in the Flow Cytometry Core at the University

of Utah. Flux lipidomics and analysis was performed at the Metabolomics Core Facility at the University of Utah. RNA-Seq library preparation and sequencing was performed by the High-Throughput Genomics Shared Resource at the Huntsman Cancer Institute. The support and resources from the Center for High Performance Computing at the University of Utah are gratefully acknowledged. This study was supported by grants from the NIH (GM115174 and GM115129 to JR, GM110755 to DW) and the Nora Eccles Treadwell Foundation (to JR), as well as HHMI (JR). Support for SMN was also provided by UMDF and ACS postdoctoral fellowships, along with T32HL007576. JAB received support from NIDDK T32DK11096601 to Wendy W Chapman and Simon J Fisher. RJD is an advisor for Agios Pharmaceuticals and is supported by HHMI, NCI (R35CA22044901) and the Once Upon a Time Foundation. AS is funded by NICHD (F32HD096786). KF is funded by R01DK107397 and R21AG063077. JEC is funded by S10OD016232, S10OD021505, and U54DK110858. JAP is funded by NIH/NIGMS grant R01 GM132129 and SPG is funded by GM97645. The content is solely the responsibility of the authors and does not necessarily represent the official views of the National Institutes of Health.

## Additional information

### Competing interests

Ralph J DeBerardinis: Reviewing editor, *eLife*. The other authors declare that no competing interests exist.

### Funding

| Funder | Grant reference number | Author |
|---|---|---|
| National Institute of General Medical Sciences | GM115174 | Jared Rutter |
| National Institute of General Medical Sciences | GM115129 | Jared Rutter |
| The Nora Eccles Treadwell Foundation | | Jared Rutter |
| Howard Hughes Medical Institute | Investigator | Ralph J DeBerardinis Jared Rutter |
| United Mitochondrial Disease Foundation | PF-15-046 | Sara M Nowinski |
| American Cancer Society | PF-18-106-01 | Sara M Nowinski |
| National Heart, Lung, and Blood Institute | T32HL007576 | Sara M Nowinski |
| National Institute of Diabetes and Digestive and Kidney Diseases | T32DK11096601 | Jordan A Berg |
| National Cancer Institute | R35CA22044901 | Ralph J DeBerardinis |
| The Once Upon a Time Foundation | | Ralph J DeBerardinis |
| Eunice Kennedy Shriver National Institute of Child Health and Human Development | F32HD096786 | Ashley Solmonson |
| National Institute of Diabetes and Digestive and Kidney Diseases | R01DK107397 | Katsuhiko Funai |
| National Institute of Diabetes and Digestive and Kidney Diseases | R21AG063077 | Katsuhiko Funai |
| Office of the Director | S10OD016232 | James E Cox |
| Office of the Director | S10OD021505 | James E Cox |

| | | |
|---|---|---|
| National Institute of Diabetes and Digestive and Kidney Diseases | U54DK110858 | James E Cox |
| National Institute of General Medical Sciences | R01GM132129 | Joao A Paulo |
| National Institute of General Medical Sciences | GM97645 | Steven P Gygi |
| National Institute of General Medical Sciences | GM110755 | Dennis R Winge |

The funders had no role in study design, data collection and interpretation, or the decision to submit the work for publication.

## Author contributions

Sara M Nowinski, Conceptualization, Formal analysis, Investigation, Visualization, Methodology, Writing - original draft, Writing - review and editing; Ashley Solmonson, Formal analysis, Investigation, Methodology, Writing - review and editing; Scott F Rusin, Investigation, Methodology; J Alan Maschek, Formal analysis, Methodology; Claire L Bensard, Sarah Fogarty, Investigation, Writing - review and editing; Mi-Young Jeong, Sandra Lettlova, Yeyun Ouyang, Investigation; Jordan A Berg, Formal analysis, Visualization; Jeffrey T Morgan, Formal analysis, Visualization, Writing - review and editing; Bradley C Naylor, Formal analysis; Joao A Paulo, James E Cox, Methodology; Katsuhiko Funai, Resources, Supervision; Steven P Gygi, Resources, Supervision, Methodology; Dennis R Winge, Supervision, Funding acquisition, Writing - review and editing; Ralph J DeBerardinis, Resources, Methodology, Writing - review and editing; Jared Rutter, Conceptualization, Resources, Supervision, Funding acquisition, Methodology, Writing - review and editing

## Author ORCIDs

Sara M Nowinski ⓘD https://orcid.org/0000-0002-4744-6101
Ashley Solmonson ⓘD http://orcid.org/0000-0001-8863-4558
Dennis R Winge ⓘD http://orcid.org/0000-0003-1160-1189
Jared Rutter ⓘD https://orcid.org/0000-0002-2710-9765

## Decision letter and Author response

Decision letter https://doi.org/10.7554/eLife.58041.sa1
Author response https://doi.org/10.7554/eLife.58041.sa2

# Additional files

## Supplementary files

• Supplementary file 1. Table of all steady-state metabolites measured in mtFAS mutants and controls. All steady-state metabolites measured via LCMS. Values shown are average fold change from mean GFP abundance.

• Supplementary file 2. Proteins associated with skeletal muscle differentiation are decreased in abundance in mtFAS mutants. Relative expression of proteins associated with skeletal muscle differentiation from SL-TMT experiment. Values are arbitrary units.

• Transparent reporting form

## Data availability

The data have been deposited to the ProteomeXchange Consortium via the PRIDE (Perez-Riverol et al., 2019) partner repository. The code used to process these data is available at Github https://github.com/j-berg/nowinski_2020 (copy archived at https://github.com/elifesciences-publications/nowinski_2020). Sequencing data have been deposited at the Gene Expression Omnibus (http://www.ncbi.nlm.nih.gov/geo) under accession number GSE148617.

The following datasets were generated:

| Author(s) | Year | Dataset title | Dataset URL | Database and Identifier |
|---|---|---|---|---|
| Nowinski Sm, Rusin SF, Paulo JA, Gygi SP, Rutter J | 2020 | Mitochondrial fatty acid synthesis coordinates oxidative metabolism in mammalian mitochondria | http://proteomecentral. proteomexchange.org/ cgi/GetDataset?ID= PXD021034 | PRIDE, PXD021034 |
| Nowinski SM, Morgan JT, Rutter J | 2020 | Mitochondrial fatty acid synthesis coordinates mitochondrial oxidative metabolism | https://www.ncbi.nlm. nih.gov/geo/query/acc. cgi?acc=GSE148617 | NCBI Gene Expression Omnibus, GSE148617 |

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
