## [Decision Letter]

**Acceptance summary:**

The paper provides novel insight into the function of mitochondria fatty acid synthesis in oxidative metabolism and will be of broad interest.

**Decision letter after peer review:**

Thank you for submitting your article "Mitochondrial fatty acid synthesis coordinates mitochondrial oxidative metabolism" for consideration by *eLife*. Your article has been reviewed by two peer reviewers, one of whom is a member of our Board of Reviewing Editors, and the evaluation has been overseen by Utpal Banerjee as the Senior Editor. The reviewers have opted to remain anonymous.

The reviewers have discussed the reviews with one another and the Reviewing Editor has drafted this decision to help you prepare a revised submission.

As the editors have judged that your manuscript is of interest, but as described below that additional experiments are required before it is published, we would like to draw your attention to changes in our revision policy that we have made in response to COVID-19 (https://elifesciences.org/articles/57162). First, because many researchers have temporarily lost access to the labs, we will give authors as much time as they need (within reason), to submit revised manuscripts. We are also offering, if you choose, to post the manuscript to bioRxiv (if it is not already there) along with this decision letter and a formal designation that the manuscript is "in revision at *eLife*". Please let us know if you would like to pursue this option. (If your work is more suitable for medRxiv, you will need to post the preprint yourself, as the mechanisms for us to do so are still in development.)

Summary:

Nowinski et al. describe a novel role for mtFAS in mitochondrial respiratory metabolism, independent of protein lipoylation, the best characterized function of the mtFAS system. mtFAS is a poorly characterized mitochondrial fatty acid synthesis pathway which is comprised of at least 6 enzymes. This pathway synthesizes an eight-carbon saturated fatty acid, which is subsequently converted into lipoic acid, an indispensable cofactor for some mitochondrial enzymes. In addition to lipoic acid precursor, mtFAS also produces longer-chain fatty acids, yet the physiological function of these mtFAS-driven longer fatty acids remain unclear. In the current manuscript, the authors show that hypomorphic mtFAS mutant cultured clonal cells exhibit a loss of ETC supercomplexes that is independent of mtFAS protein lipoylation activity.

Although the results are of interest, the are several issues related to study design and level of mechanistic insight that need to be addressed. How the myFAS pathway regulates the ECT complexes assembly and stability required for optimal mitochondrial activity and how it controls myoblast differentiation are unclear. Without such data, this study will have more limited impact.

Essential revisions:

1) A Crispr/Cas9-based strategy is used on skeletal myoblasts to mutate mtFAS genes, followed by the generation of clonal cell lines. This strategy was only partially successful, as the authors failed to identify a single null mutant for any of the genes targeted. Instead, the authors opted for a detailed comparison of distinct clonal cell lines with incomplete target ablation. This is technically challenging due to possible interclonal heterogeneity and long-term compensatory mechanisms. The authors should either perform rescue experiments within clonal cell lines or consider a short-term knockdown approach in polyclonal myoblast cell lines.

2) It remains unclear how mtFAS supports ETC function, either by producing free metabolites or acyl-ACP. To demonstrate acylation-dependency, the authors could evaluate if a mutant acylation-free ACP1 can interact with LYRM proteins (NDUFA6 and LYRM4). In addition, the authors should examine mtFAS-dependent physical interaction of ACP and LYRM proteins in wild-type and mtFAS mutant cells.

3) In Figure 5, the authors suggest that the defective ETC assembly and reductive carboxylation are independent of protein lipoylation by using Lipt_1/2_ mutant cells. This is the pivotal experiment in this manuscript. The authors should measure MCAT, OXSM and MECR protein levels in Lipt_1/2_ mutant cells.

---

## [Author Response]

Essential revisions:1) A Crispr/Cas9-based strategy is used on skeletal myoblasts to mutate mtFAS genes, followed by the generation of clonal cell lines. This strategy was only partially successful, as the authors failed to identify a single null mutant for any of the genes targeted. Instead, the authors opted for a detailed comparison of distinct clonal cell lines with incomplete target ablation. This is technically challenging due to possible interclonal heterogeneity and long-term compensatory mechanisms. The authors should either perform rescue experiments within clonal cell lines or consider a short-term knockdown approach in polyclonal myoblast cell lines.

We thank the reviewers for this suggestion. In response, we have now constructed genetic rescue clonal lines of Mcat, Oxsm, and Mecr mutants by stably expressing retroviral constructs for each gene in the appropriate mutant. As expected, we show that re-expression of the relevant mtFAS gene in the cognate mutant cell population rescues growth in both glucose and galactose, which requires mitochondrial respiration (new Figure 1C and Figure 1—figure supplement 1 C and D). We also show that this maneuver also rescues lipoylation, SDHB stability, and ETC complex stability/assembly via BN-PAGE (Figure 2E, 2F, Figure 2—figure supplement 1 D and E).

2) It remains unclear how mtFAS supports ETC function, either by producing free metabolites or acyl-ACP. To demonstrate acylation-dependency, the authors could evaluate if a mutant acylation-free ACP1 can interact with LYRM proteins (NDUFA6 and LYRM4). In addition, the authors should examine mtFAS-dependent physical interaction of ACP and LYRM proteins in wild-type and mtFAS mutant cells.

We share the reviewers’ intense interest in this question. Unfortunately, while it may seem approachable on the surface, a number of issues detailed below make it an incredibly difficult question to answer. However, evidence from a number of sources support our conclusion that ACP acylation and LYRM binding is the key factor in regulation of ETC assembly.

Acyl chains are covalently linked to ACP via a 4’-phosphopantetheine (4’-PP) co-factor that is attached to a conserved serine residue. Structural studies from several labs have shown that the 4’-PP co-factor participates in the physical interactions with LYRM proteins, including LYRM4 and NDUFA6 (Cory et al., 2017; Fiedorczuk et al., 2016). These studies and others have shown that mutant ACP loses physical interaction with LYRM4 (Cory et al., 2017; Majmudar et al., 2019; Van Vranken et al., 2016; Van Vranken et al., 2018). However, to accurately assess the requirement of ACP acylation one cannot use a mutant (lacking the serine residue), acylation-free ACP because this construct also lacks the 4’-PP. These studies cannot separate the effects of loss of 4’-PP from the effects of loss of acylation. Therefore, we did not repeat the numerous studies in the literature using a mutant ACP at this time.

The second question raised by the reviewers, regarding the mtFAS-dependent physical interaction of ACP and the LYRM proteins LYRM4 and NDUFA6 in wild type and mtFAS mutant cells, is also essentially impossible to address because, as demonstrated by our proteomics data, NDUFA6 and LYRM4 are not stable in mtFAS mutant cells. We have now added western blot data confirming the decreased abundance of LYRM4 that we previously observed in our proteomics study. These data demonstrate that it is undetectable by western blot in mtFAS mutant cell lines (Figure 3H). Although we purchased 3 separate commercially available NDUFA6 antibodies, none gave us detectable signal of a reasonably sized protein (one did give a band at 75kDa, however NDUFA6 has a predicted molecular weight of 18kDa).

We have previously observed a perfect correlation between LYRM stability and ACP interaction in yeast mtFAS mutants (Van Vranken et al., 2018). In other words, when the LYRM protein is stable, interaction with ACP is maintained in the absence of acylation, thus we speculate from these data that NDUFA6 and LYRM4 require interaction with acylated ACP for their stability. However, when the LYRM protein is not stable in the absence of acylation, this makes it difficult to assess changes in physical interaction via co-immunoprecipitation, because the LYRM proteins are greatly decreased in abundance in the input samples (Van Vranken et al., 2018). Nevertheless, we performed co-immunoprecipitation experiments with a tagged ACP and endogenous LYRM4 in wildtype and mtFAS mutant cells as requested by reviewers. As you can see in Author response image 1, LYRM4 co-immunoprecipitates with ACP specifically in wild type cells expressing the tagged ACP, however the experiment is uninterpretable because LYRM4 is absent from the Oxsm mutant input, in agreement with our other data. Because this experiment is not interpretable as to the causal mechanistic question, we feel it should not be included in the manuscript.

We were hesitant to overexpress tagged LYRM4 and NDUFA6 constructs because we feared that this might overcome decreased binding affinity in the absence of acylation. However, it appears that in fact even overexpression of these LYRM proteins does not create stably expressed tagged protein. Despite achieving 30% transfection efficiency with peGFP in our wild type and mtFAS mutant C2C12 lines, we were unable to detect the tagged LYRM proteins (expressed from the CMV promoter) in either input or immunoprecipitated lysates. We then turned to 293T as a more robustly transfectable system to verify the expression and interaction of tagged LYRM4, even though we lack a mtFAS mutant 293T cell line. Even in wild type 293T, overexpression of tagged LYRM4 proved very difficult, and was only faintly visible when co-immunoprecipitated with ACP (Author response image 2). We believe this is likely because overexpression of these LYRM proteins in the absence of their other binding partners (NFS1 and ISCU in the case of LYRM4, and complex I in the case of NDUFA6) does not result in the creation of stable protein, an effect that has been seen for numerous other electron transport chain subunits. If the reviewers feel that inclusion of these Co-IP data in 293T improve the manuscript, we are happy to include this data in Figure 3—figure supplement 1.

**Author response image 2. respfig2:** 

3) In Figure 5, the authors suggest that the defective ETC assembly and reductive carboxylation are independent of protein lipoylation by using Lipt_1/2_ mutant cells. This is the pivotal experiment in this manuscript. The authors should measure MCAT, OXSM and MECR protein levels in Lipt_1/2_ mutant cells.

We have added this data as the new Figure 5—figure supplement 1. Consistent with our hypothesis, Lipt1 mutants show no difference or slightly increased expression of MCAT, OXSM, and MECR proteins. Therefore, the loss of lipoylation in Lipt1 mutant cells can’t be ascribed to decreased expression of these mtFAS enzymes.